# Wnt3 distribution in the zebrafish brain is determined by expression, diffusion and multiple molecular interactions

**Sapthaswaran Veerapathiran[1,2], Cathleen Teh[1,2], Shiwen Zhu[1,2], Indira Kartigayen[1,2], Vladimir Korzh[3], Paul T Matsudaira[1,2], Thorsten Wohland[1,2,4]***

[1]Department of Biological Sciences, National University of Singapore, Singapore, Singapore; [2]Center for BioImaging Sciences, National University of Singapore, Singapore, Singapore; [3]International Institute of Molecular and Cell Biology in Warsaw, Warsaw, Poland; [4]Department of Chemistry, National University of Singapore, Singapore, Singapore

**Abstract** Wnt3 proteins are lipidated and glycosylated signaling molecules that play an important role in zebrafish neural patterning and brain development. However, the transport mechanism of lipid-modified Wnts through the hydrophilic extracellular environment for long-range action remains unresolved. Here we determine how Wnt3 accomplishes long-range distribution in the zebrafish brain. First, we characterize the Wnt3-producing source and Wnt3-receiving target regions. Subsequently, we analyze Wnt3 mobility at different length scales by fluorescence correlation spectroscopy and fluorescence recovery after photobleaching. We demonstrate that Wnt3 spreads extracellularly and interacts with heparan sulfate proteoglycans (HSPG). We then determine the binding affinity of Wnt3 to its receptor, Frizzled1 (Fzd1), using fluorescence cross-correlation spectroscopy and show that the co-receptor, low-density lipoprotein receptor-related protein 5 (Lrp5), is required for Wnt3-Fzd1 interaction. Our results are consistent with the extracellular distribution of Wnt3 by a diffusive mechanism that is modified by tissue morphology, interactions with HSPG, and Lrp5-mediated receptor binding, to regulate zebrafish brain development.

*For correspondence:
twohland@nus.edu.sg

**Competing interests:** The authors declare that no competing interests exist.

## Introduction

Wnt proteins represent a family of secreted signaling glycoproteins having multiple functions in embryonic development such as specification of the vertebrate axis, embryonic induction, maintenance of cell potency, cell fate determination, cell migration, cell division, and apoptosis, to name a few (*Clevers and Nusse, 2012*; *Hikasa and Sokol, 2013*; *Logan and Nusse, 2004*; *Moon et al., 2002*). So far, 13 *wnt* gene subfamilies have been identified, although the number of *wnt* genes differs between species (*Schubert and Holland, 2013*). Wnts are generally 350–400 amino acids in length (molecular weight of ~40 kDa), with highly conserved cysteine residues. Wnts are hydrophobic and water insoluble due to their post-translational lipidation in the endoplasmic reticulum (ER) (*Mikels and Nusse, 2006*). Porcupine (Porc), an O-acyltransferase localized on the membrane of the ER, catalyzes the acylation of Wnts and provides Wnts hydrophobic characteristics (*Herr and Basler, 2012*). The acylation facilitates the interaction of Wnts with Wntless, a transmembrane protein that shuttles Wnts to the plasma membrane (*Galli et al., 2007*). From the plasma membrane, they are secreted and transported to Wnt-receiving cells. Hence, the acylation of Wnts is a critical step for their trafficking, secretion, and activity (*Coudreuse and Korswagen, 2007*).

However, the addition of lipid moieties makes the long-range diffusion of Wnts in the aqueous extracellular matrix problematic. Several transport mechanisms were proposed to explain how Wnts

navigate the aqueous environment to achieve long-range action (*Routledge and Scholpp, 2019*). Facilitated shuttling of Wnts by chaperone proteins is a commonly reported mode of distribution. Here Wnt-binding proteins such as secreted Frizzled-related proteins (sFRPs) (*Esteve et al., 2011*; *Mii and Taira, 2009*), secreted Wg-interacting Molecule (Swim) (*Mulligan et al., 2012*), or afamin (*Mihara et al., 2016*) shield the hydrophobic regions of Wnts and provide stability in the aqueous environment. Similarly, hydrophobic Wnt ligands could be packaged inside exosomes and lipoprotein particles, which enable their extracellular movement (*Greco et al., 2001*; *Neumann et al., 2009*; *Panáková et al., 2005*). Additionally, heparan sulfate proteoglycans (HSPG) present in the extracellular matrix serve as binding sites for several signaling molecules, including Wnts (*Kirkpatrick and Selleck, 2007*). HSPG maintains the solubility of Wnt ligands and prevents their aggregation in the aqueous extracellular matrix, thereby enhancing their signaling range (*Fuerer et al., 2010*; *Mii et al., 2017*). Further evidence suggests that HSPG in coordination with Wnts are pivotal in regulating gastrulation, neurulation, and axis formation during embryonic development (*Ohkawara et al., 2003*; *Saied-Santiago et al., 2017*; *Tao et al., 2005*; *Topczewski et al., 2001*). On the other hand, it was also recently noticed that certain Wnts could be deacylated by Notum, a secreted deacylase, and maintain reduced signaling activity (*Speer et al., 2019*). Besides the extracellular transport mechanism, certain Wnt proteins may also reach their target tissues through filopodial extensions called cytonemes, as seen for Wnt2b in *Xenopus* (*Holzer et al., 2012*), Wg in Drosophila (*Huang and Kornberg, 2015*), and Wnt8a in zebrafish embryos (*Mattes et al., 2018*; *Stanganello et al., 2015*).

Finally, when Wnts reach their target tissues, they bind to their target receptors and elicit a signaling cascade. To date, Wnts are known to interact with more than 15 receptor and co-receptor protein families (*Niehrs, 2012*), of which the Frizzled (Fzd) receptor super-family is the most commonly investigated. Fzd proteins are categorized under the Class-F super-family of G-protein coupled receptors. The super-family comprises 10 Fzd receptors (Fzd1-Fzd10) and Smoothened (SMO) (*Schulte and Wright, 2018*), all with a seven-pass transmembrane domain and a highly conserved cysteine-rich domain (CRD) (*Hsieh et al., 1999*; *Wu and Nusse, 2002*). Structural studies revealed that the low-density lipoprotein receptor-related protein (Lrp-5/6) acts as a co-receptor and is involved with the Wnt-Fzd complex (*Chu et al., 2013*; *Hirai et al., 2019*; *Janda et al., 2012*). The Wnt-Fzd-Lrp complex inhibits the negative regulator destruction complex and stabilizes the Wnt signaling transducer β-catenin, which allows the transcription of genes regulating embryonic development and patterning (*Bilic et al., 2007*).

Wnt3 proteins, a subset of the Wnt family, are instrumental in the development of the nervous system, vascular system, limb formation, and vertebrate axis formation (*Anne et al., 2013*; *Bulfone et al., 1993*; *Clements et al., 2009*; *Garriock et al., 2007*; *Liu et al., 1999*). In zebrafish, Wnt3 directs neural stem cell proliferation and differentiation, making it indispensable for brain development (*Clements et al., 2009*). Our group showed that in zebrafish embryos, Wnt3 associates with domains on the membrane (*Azbazdar et al., 2019*; *Ng et al., 2016*; *Sezgin et al., 2017*). Blocking the activity of Porc and thus reducing Wnt acylation resulted in reduced domain confinement and defective brain development in zebrafish embryos (*Ng et al., 2016*; *Teh et al., 2015*). The understanding of the Wnt3 action mechanism in zebrafish brain development, therefore, requires identifying its source regions, determining its mode of transport, demarcating receiving target tissues, and measuring Wnt3-receptor interactions.

In this study, we first mapped the source and target regions of Wnt3 in the zebrafish brain by comparing the expression of a transgenic line expressing functional Wnt3EGFP, with a reporter line expressing an inner plasma membrane targeting sequence tagged with mApple (PMTmApple). The expression in both lines is regulated by a 4 kb *wnt3* promoter that contains most of the regulatory elements and reports the spatiotemporal expression of *wnt3* (*Teh et al., 2015*). Wnt3EGFP spreads from where it is produced, while PMTmApple remains attached to the inner membrane leaflet of the producing cells. Hence, by analyzing the expression patterns of Wnt3EGFP and PMTmApple, we were able to classify the dorsal regions of the cerebellum (dCe), the midbrain–hindbrain boundary (MHB), the brain midline (midbrain roof plate [mRP] and floor plate [FP]) and the epithalamus (Epi) as source regions of Wnt3, and the optic tectum (OT) and ventral regions of the cerebellum (vCe) as distal target regions. Subsequently, we probed how Wnt3 is distributed from the source to the target regions of the zebrafish brain by measuring its in vivo dynamics using fluorescence correlation spectroscopy (FCS) and fluorescence recovery after photobleaching (FRAP). FCS is a single molecule

sensitive technique that statistically analyzes the intensity fluctuations in a small observation volume (~femtoliter scale) to generate an autocorrelation function (ACF), from which the diffusion coefficient and the concentration of the fluorescent molecules in the observation volume are accurately evaluated (*Enderlein et al., 2005*; *Kim et al., 2007*; *Krichevsky and Bonnet, 2002*; *Magde et al., 1974*). FRAP, on the contrary, is an ensemble technique that measures the dynamics of the fluorescent molecules in a large region of interest (~micrometer scale) based on the recovery of the fluorescence intensity in an irreversibly photobleached region (*Klonis et al., 2002*; *Koppel et al., 1976*). FCS and FRAP both measure molecular mobilities and have been shown to provide consistent results in vitro (*Macháň et al., 2016*). However, the global diffusion coefficient ($D_{global}$) measured from FRAP is significantly lower in comparison to its local diffusion coefficient ($D_{local}$) obtained from FCS for several morphogens in organisms (*Müller et al., 2012*; *Müller et al., 2013*). Here we study the local and global diffusion of Wnt3EGFP and identify the possible factors that leads to the discrepancy between its $D_{local}$ and $D_{global}$.

Lastly, we monitored the in vivo interaction of Wnt3 with Fzd1, a potential target receptor, using fluorescence cross-correlation spectroscopy (FCCS) and calculated their binding affinity. In FCCS, the intensity fluctuations of two interacting molecules tagged with spectrally different fluorophores in an observation volume are cross-correlated, and the dissociation constant between the interacting species can be quantitatively determined in live cells and organisms (*Ries et al., 2009*; *Schwille et al., 1997*; *Shi et al., 2009*; *Sudhaharan et al., 2009*; *Wang et al., 2016*; *Yavas et al., 2016*). We observed that the co-receptor Lrp5 is essential for the interaction of Wnt3 with Fzd1. Our findings show that Wnt3EGFP spreads from its source to distal target regions by extracellular diffusion and the difference in local and global diffusion is due to tissue geometry, interactions with HSPG, and its receptors.

## Results

### Identifying the source and target regions for Wnt3

In order to identify the source and target regions of Wnt3, we used two transgenic lines: Tg(−4.0*wnt3*:Wnt3EGFP) and Tg(−4.0*wnt3*:PMTmApple). Tg(−4.0*wnt3*:Wnt3EGFP) is a functionally active Wnt3EGFP-expressing line driven by a 4 kb *wnt3* promoter that positively regulates tissue growth in midbrain, MHB, and cerebellum (*Teh et al., 2015*). Tg(−4.0*wnt3*:PMTmApple) is a reporter line driven by the same 4 kb *wnt3* promoter, expressing PMTmApple. Since the 4 kb *wnt3* promoter contains most of the regulatory elements (*Teh et al., 2015*), Tg(−4.0*wnt3*:PMTmApple) is a faithful reporter of Wnt3 expression, which marks the plasma membrane of the Wnt3-producing cells. However, the localization of PMTmApple is restricted to its source cells, as it remains tethered to the inner leaflet of the plasma membrane. In contrast, the distribution pattern of Wnt3EGFP in Tg(−4.0*wnt3*:Wnt3EGFP) spans a broader range compared to PMTmApple in Tg(−4.0*wnt3*:PMTmApple), implying that Wnt3EGFP is transported from the source regions where it is produced to its distal target regions (*Figure 1*). The overlap in the expression of the two lines, therefore, identifies the source regions, and the difference demarcates the distal target regions.

The two transgenic lines were crossed [Tg(−4.0*wnt3*:Wnt3EGFP) × Tg(−4.0*wnt3*:PMTmApple)] and the expression of Wnt3EGFP and PMTmApple were sequentially recorded using a confocal microscope in their respective wavelength channels. First, the obtained image stacks were segmented using an automatic threshold algorithm (*Zhu et al., 2016*), and the colocalization of each pixel was evaluated based on the intensity correlation analysis (ICA), the distance weight, and intensity weight (*Li et al., 2004*; *Zhu et al., 2016*) to generate a pair of masks for the colocalized and non-colocalized pixels. Subsequently, color-coded heat maps were generated, indicating the contribution of each pixel to the overall colocalization at 24 and 48 hpf (*Videos 1* and *2*). Finally, using the colocalized and non-colocalized masks, volumetric images were constructed to distinguish the source and target regions of Wnt3 respectively. At 24 hpf, the source regions were MHB, dCe, and Epi, whereas the distal target regions were vCe and OT (*Figure 2* and *Video 3*). The source regions at 48 hpf were mRP, FP, MHB, dCe, Epi, and some parts of the dorso-lateral optic tectum (dOT), while the distal target regions were vCe and ventral optic tectum (vOT) (*Figure 3* and *Video 4*). With the source and target regions defined, we next quantified the dynamics of Wnt3EGFP and examined the mode of dispersal of Wnt3EGFP from its source to the distal target regions.

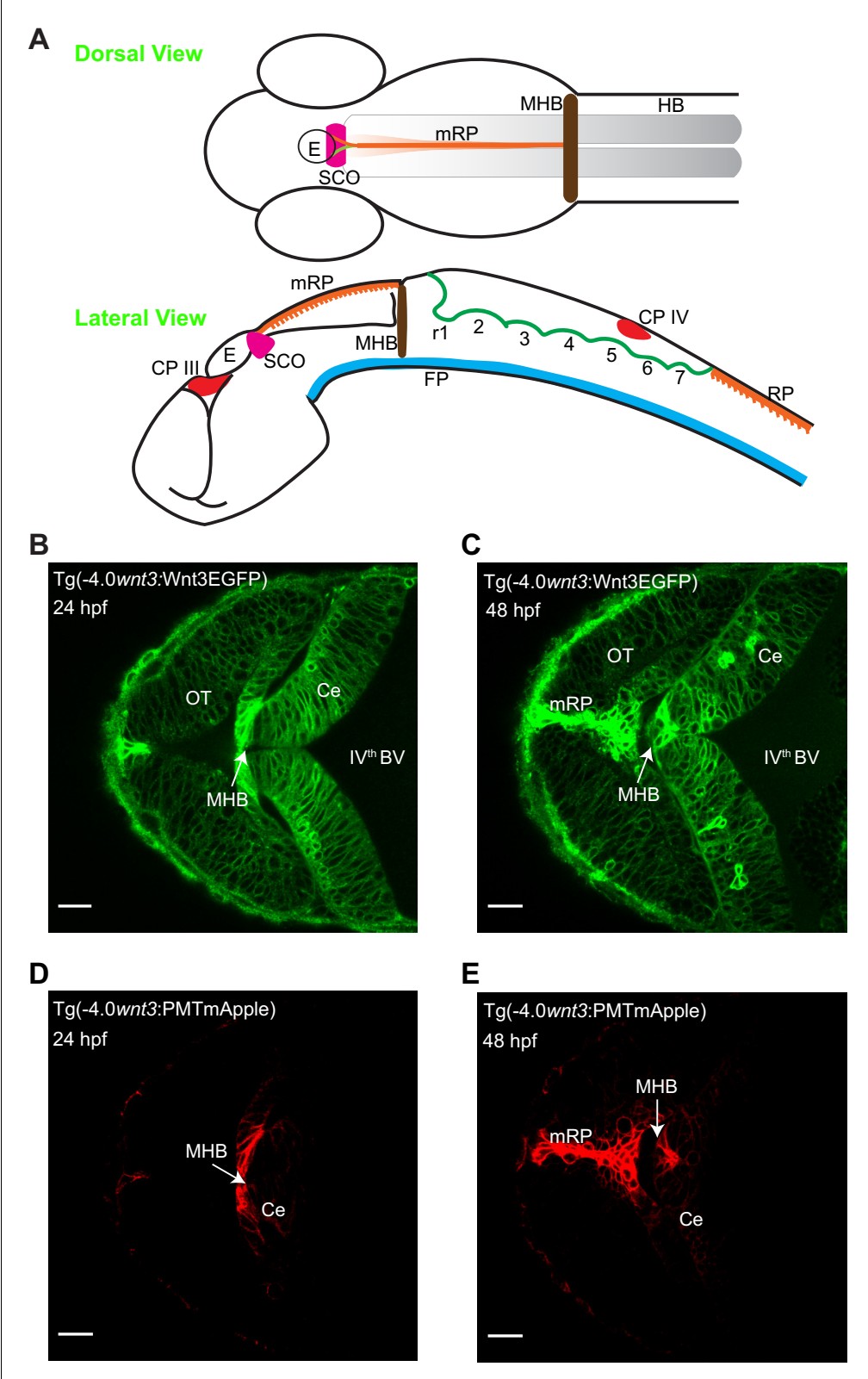

**Figure 1.** Spatiotemporal expression of *wnt3* promoter-driven Wnt3EGFP and PMTmApple. (A) Schematic illustration of the brain of a zebrafish embryo (dorsal view and lateral view). Expression profile of Wnt3EGFP in Tg(−4.0*wnt3*:Wnt3EGFP) line at (B) 24 hpf and (C) 48hpf. Expression profile of PMTmApple in Tg(−4.0*wnt3*:PMTmApple) line at (D) 24 hpf and (E) 48hpf. BV, brain ventricle; Ce, cerebellum; CP, choroid plexus; E, epiphysis; FP, floor

*Figure 1 continued on next page*

*Figure 1 continued*

plate; HB, hindbrain; MHB, midbrain–hindbrain boundary; mRP, midbrain roof plate; OT, optic tectum; r, rhombomere; RP, roof plate (spinal cord); SCO, sub-commissural organ. SCO and E are regions of the epithalamus (Epi). Images orientation: anterior to the left. Scale bar 30 μm.

## Characterizing the in vivo dynamics of Wnt3EGFP

Wnt3EGFP was detected in the fourth brain ventricle (BV), and along the cell borders in midbrain, MHB, and hindbrain (*Figure 4A*). In the BV, Wnt3EGFP diffuses freely with a diffusion coefficient of 54.6 ± 11.3 μm$^2$/s. In addition, a slow diffusing fraction with a diffusion coefficient of 4.8 ± 3.4 μm$^2$/s was also detected in the BV. The ACF were fit using a 3D-2particle-1triplet model for Wnt3EGFP in the BV, as determined by Bayes inference-based model selection (*Sun et al., 2015*; *Teh et al., 2015*) (see Materials and methods). In the cerebellum, MHB, and optic tectum, the cells are densely packed at 24 hpf and 48hpf, and there is no apparent extracellular space resolved within the limits of our microscopes (~200 nm). It is thus not possible to determine from imaging alone whether Wnt3EGFP is present in the interstitial spaces. We therefore use an indirect approach and measure the molecular mobility of Wnt3EGFP at the borders between neighboring cells using FCS (*Figure 4B*). As diffusion coefficients on membranes and in aqueous solution differ by at least one order of magnitude if not more, they can be easily distinguished, and the presence of a freely diffusible species can be identified. For FCS measurements along the cell borders, we used a 2D-2particle-1triplet fit model (*Equation 7* in Materials and methods) . The fact that data can be fit with a 2D model most likely indicates that Wnt3EGFP either diffuses on the membrane or in the narrow interstitial spaces that have a very small extent (<200 nm) compared to the axial extent of the confocal volume (~1 μm). The two detected diffusive components comprise a slow component, which was the dominant fraction ($F_{slow}$ ~0.6 ± 0.05), with a diffusion coefficient ($D_{slow}$) of 0.6 ± 0.3 μm$^2$/s and a fast component with a diffusion coefficient ($D_{fast}$) of 27.6 ± 3.9 μm$^2$/s (*Figure 4E*). The diffusion coefficients of proteins range roughly between 0.1 and 2 μm$^2$/s in cell membranes and depend on their localization and interactions within the membrane. As shown in our previous studies, Wnt3 is associated with lipid domains on the membrane and is influenced by various interventions that change the lipid content of membranes in cells and in vivo (*Azbazdar et al., 2019*; *Ng et al., 2016*; *Sezgin et al., 2017*). Therefore, the slow component likely represents the fraction of Wnt3EGFP on the membrane. On the other hand, the fast component is too fast to be attributed to diffusion within the plasma membrane. Note, however, that we cannot unambiguously assign the fast diffusion coefficient to Wnt3EGFP in the interstitial spaces. The confocal volume for FCS measurements on the membrane also spans a portion of the intracellular cytosol. Hence, a fraction of Wnt3EGFP within the cytosol could have contributed to the fast diffusion. Therefore, to check whether Wnt3EGFP diffuses in the extracellular spaces, we tested whether the fast diffusion coefficient is susceptible to changes in the interstitial spaces.

## Wnt3EGFP spreads extracellularly in the interstitial spaces

As Wnts are highly hydrophobic molecules, they tend to aggregate after being secreted into the extracellular milieu, which would limit them to autocrine and juxtacrine signaling (*Fuerer et al., 2010*). However, the expression of Wnt3EGFP in Tg(−4.0*wnt3*:Wnt3EGFP) was detected at a distance (~50–150 μm) from the recognized source regions, implying long-range travel. Hence, we examined how Wnt3EGFP spreads across the zebrafish brain, and whether it chooses the extracellular route. Since the cells are tightly packed at late stages (after 24 hpf) of the zebrafish embryo, we first verified the existence of the interstitial spaces at these late stages. We injected secreted EGFP mRNA (secEGFP), the

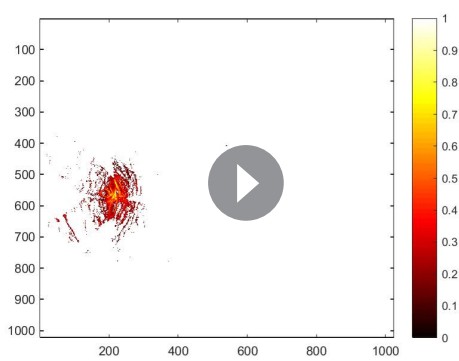

**Video 1.** Colocalization of Wnt3EGFP and PMTmApple in Tg(−4.0wnt3:Wnt3EGFP) × Tg(−4.0wnt3: PMTmApple) at 24 hpf. https://elifesciences.org/articles/59489#video1

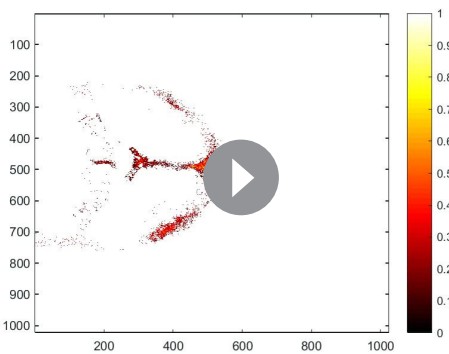

**Video 2.** Colocalization of Wnt3EGFP and PMTmApple in Tg(−4.0wnt3:Wnt3EGFP) × Tg(−4.0wnt3: PMTmApple) at 48 hpf.

https://elifesciences.org/articles/59489#video2

secretory peptide of Fibroblast growth factor 8a (Fgf8a) tagged with EGFP, at the one-cell stage and imaged the zebrafish brain at 48 hpf. The secEGFP construct is targeted for extracellular secretion after its translation in the cytoplasm and it is a marker for interstitial spaces. We observed the expression of secEGFP along the cell boundaries of the zebrafish brain even at tightly packed stages (≥24 hpf) and in the BV (*Figure 4C*). When the dynamics for secEGFP was measured using FCS, we obtained a D of $57.9 \pm 14.4$ µm$^2$/s along the cell boundaries and a D of $87.5 \pm 11.3$ µm$^2$/s in the BV (*Figure 4D, E*). These values are independent of whether secEGFP is injected as mRNA or whether it is expressed under the control of the 4 kb *wnt3* promoter (*Figure 4—figure supplement 1*). As secEGFP does not bind to the cell membrane, this indicates its diffusion in the extracellular spaces, consistent with the fast diffusion coefficient measured (*Müller et al., 2012*; *Müller et al., 2013*).

As mentioned above, we were unable to unambiguously assign Wnt3EGFP diffusion to its presence in interstitial spaces. Thus, we evaluated the effects of HSPG, a cell surface, and extracellular matrix protein which should influence only molecules in interstitial spaces, on the dynamics of Wnt3EGFP. Since the interactions of Wnts with HSPG and the significance of HSPG in the activity of Wnts are well established (*Fuerer et al., 2010*; *Kirkpatrick and Selleck, 2007*; *Mii et al., 2017*), we treated the Tg(−4.0*wnt3*:Wnt3EGFP) embryos with heparinase to disrupt HSPG and measured the dynamics of Wnt3EGFP. Injecting heparinase at the one-cell stage showed impaired gastrulation, so heparinase along with a high molecular weight fluorescent dextran (70,000 MW Dextran-TRITC) was co-injected in the BV of 48 hpf Wnt3EGFP expressing embryos. Since the presence of fluorescent dextran was detected along cell boundaries of the cerebellum and OT, we inferred that heparinase (~42 kDa) also diffused into the interstitial spaces from the BV (*Table 1—source data 1*). Confocal FCS measurements revealed that while $D_{slow}$ of Wnt3EGFP for heparinase treated and untreated embryos remained the same, $D_{fast}$ for heparinase treated embryos was almost twofold higher ($D_{fast} = 43.4 \pm 7.6$ µm$^2$/s) in comparison with untreated embryos ($D_{fast} = 24.7 \pm 4.8$ µm$^2$/s) (*Table 1*). Additionally, we also treated Wnt3EGFP embryos with surfen, a quinolone-based derivative that exhibits heparin neutralizing activity and antagonizes heparan sulfate–protein interactions (*Naini et al., 2018*). Similar to heparinase treatment, surfen-treated Wnt3EGFP embryos had a higher $D_{fast}$ of $42.0 \pm 6.9$ µm$^2$/s whereas $D_{slow}$ remained unchanged ($0.5 \pm 0.3$ µm$^2$/s) (*Table 1*). As Wnts are known to travel in the extracellular spaces by constant binding and unbinding to HSPG (*Yan and Lin, 2009*), the change in diffusion coefficient upon HSPG cleavage is only seen for the fast component in the extracellular spaces and not the slow component.

As controls, we measured the effects of heparinase and surfen treatment on the diffusion of secEGFP and LynEGFP (a non-functional membrane tethered tyrosine kinase) as they are not known to interact with HSPG. When secEGFP embryos were treated with heparinase or surfen, we observed no changes in D compared to the untreated embryos implying that increase in $D_{fast}$ for Wnt3EGFP was not due to clearing up of the interstitial spaces (*Table 1*). For LynEGFP in Tg(−8.0*cldnB*: LynEGFP), we obtained a slow component with a $D_{slow}$ of $2.2 \pm 0.6$ µm$^2$/s and a fast component with a $D_{fast}$ of $39.1 \pm 11.2$ µm$^2$/s. As the $D_{slow}$ is within the range of diffusion coefficients observed for proteins on the membrane (0.1–2 µm$^2$/s), the $D_{slow}$ corresponds to the membrane diffusing component while $D_{fast}$ represents a putative cytosolic fraction. When LynEGFP embryos were treated with heparinase or surfen, we did not observe any changes in $D_{fast}$ or $D_{slow}$, confirming that neither membrane nor cytosolic diffusion is influenced upon HSPG disruption (*Table 1*). Since we observe a twofold increase in $D_{fast}$ only for Wnt3EGFP, but not for secEGFP and LynEGFP, upon HSPG cleavage our results suggest an extracellular distribution of Wnt3EGFP regulated by constant binding and unbinding with HSPG.

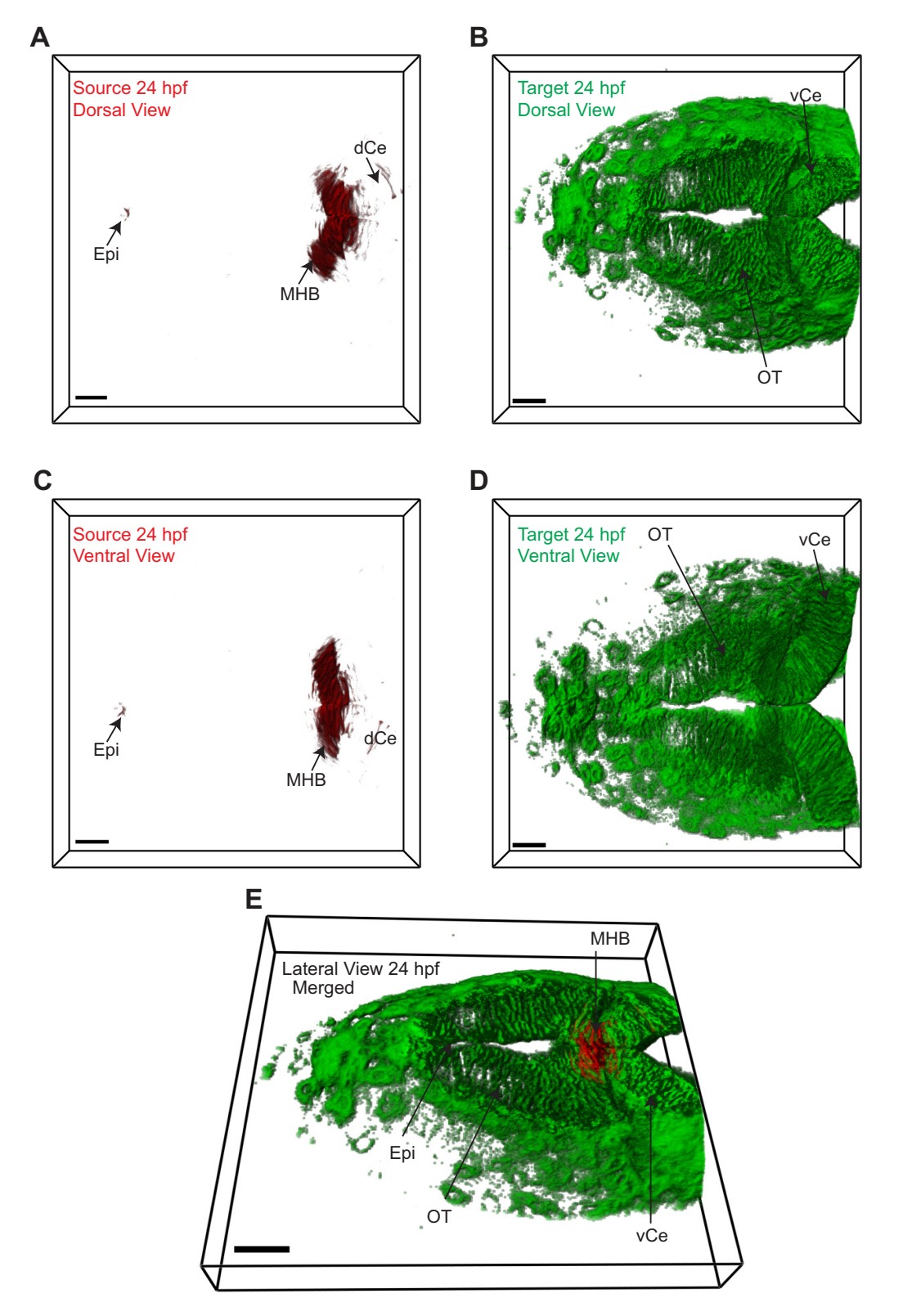

**Figure 2.** Wnt3 source and target regions at 24 hpf. 3D dorsal projection of Wnt3 (**A**) source regions at 24 hpf and (**B**) target regions at 24 hpf (top view). 3D ventral projection of Wnt3 (**C**) source regions at 24 hpf and (**D**) target regions at 24 hpf (bottom view). (**E**) 3D projection of Wnt3 source and target regions at 24 hpf (lateral view). See *Video 3* for a detailed view. dCe, dorsal regions of cerebellum; Epi, epithalamus; MHB, midbrain–hindbrain boundary; OT, optic tectum; vCe, ventral regions of cerebellum. Images orientation: anterior to the left. Scale bar 30 μm.

*Figure 2 continued on next page*

*Figure 2 continued*

The online version of this article includes the following figure supplement(s) for figure 2:

**Figure supplement 1.** Fluorescence correlation spectroscopy (FCS) measurements of Wnt3EGFP and PMTmApple in the Wnt3 target regions at 24 hpf.
**Figure supplement 2.** Expression of EGFP in Tg(−4.0*wnt3*:EGFP) at 24 hpf.
**Figure supplement 3.** Expression of *wnt3* transcripts and downstream wnt signaling transcription factor at 24 hpf.

To substantiate our results, we studied the global diffusion of Wnt3EGFP using FRAP. As FRAP measures mobility over a range of several cell diameters, it is an ideal tool to investigate whether Wnt3 can diffuse extracellularly or by other slower cellular mechanisms. We irreversibly photobleached a region of the zebrafish brain in Tg(−4.0*wnt3*:Wnt3EGFP) embryos, and observed the rate of recovery in the photobleached region. On analyzing the FRAP curve for Wnt3EGFP, two components with different time constants were obtained: a fast component with a time constant ($\tau_{fast}$) of 5–8 min and a slow component with a time constant $\tau_{slow}$ >40 min. The fast component likely denotes the recovery due to diffusion in the extracellular spaces as cell-based mechanisms would have much slower recovery rates. The slow component likely represents the recovery due to production of the fluorescent protein as the recovery time corresponds to the time taken for translation and maturation of the fluorophore. The mobile fraction ($F_m$) of Wnt3EGFP evaluated from FRAP was 0.3–0.4 with an apparent global diffusion coefficient ($D_{global}$) of $0.5 \pm 0.2$ $\mu m^2$/s (*Figure 5*), almost 40–100 times slower when compared to the local diffusion coefficient obtained from FCS ($D_{local} = 24.7 \pm 4.8$ $\mu m^2$/s). The recovery rates for Wnt3EGFP were similar at the source (*Figure 5E*) and at distal target regions (*Figure 5—figure supplement 1*). When HSPG were disrupted, faster recovery was observed ($\tau_{fast}$ ~2.5 to 3 min) with an increase of a factor ~2–3 in $D_{global}$ ($1.3 \pm 0.5$ $\mu m^2$/s) (*Figure 5—figure supplement 2*), indicating that Wnt3EGFP diffusion is slowed down by transient interactions with HSPG in the extracellular spaces. When FRAP was performed for secEGFP in the same region of the zebrafish brain, secEGFP showed rapid recovery with a $\tau_{fast}$ of ~30 s and a $D_{global}$ of $13 \pm 4$ $\mu m^2$/s ($F_m$ of 0.7–0.9), almost three to five times slower than its $D_{local}$ measured from FCS (55–60 $\mu m^2$/s) (*Figure 5—figure supplement 3*). On the other hand, PMTmApple showed no recovery within the same measurement time of the experiment (40 min) (*Figure 5—figure supplement 4*). Although the source regions continuously produce PMTmApple, the generation of novel PMTmApple involves transcription, translation, and post-translational chromophore maturation (maturation time for mApple is ~30 min). Since PMTmApple is tethered to the cell membrane, no recovery is observed after photobleaching of PMTmApple before 30 min. Overall, the FRAP results substantiate our hypothesis that Wnt3EGFP achieves long-range distribution in the zebrafish brain by diffusing in the extracellular spaces with constant binding and unbinding to HSPG.

## The in vivo interactions of Wnt3 with Fzd1 receptor depend on the expression of *lrp5* co-receptor

Apart from the interactions of signaling molecules with the extracellular matrix proteins, the transient trapping of ligands by their receptors also shapes their distribution profile (*Müller et al., 2013*). For instance, the transient binding of Nodals to their receptor Acvr2b and co-receptor Oep (*Lord et al., 2019*; *Wang et al., 2016*), Hedgehog to the 12-transmembrane protein Dispatched (*Callejo et al., 2011*), and Wingless to the Fzd receptor (*Baeg et al., 2004*) influence their respective distributions and gradient kinetics. Hence, it is critical to evaluate the binding affinity of Wnt3 with its target receptors to understand its signaling range and action. Although the binding affinities for different Wnt ligands and Fzd receptors were quantified, they were limited to biochemical

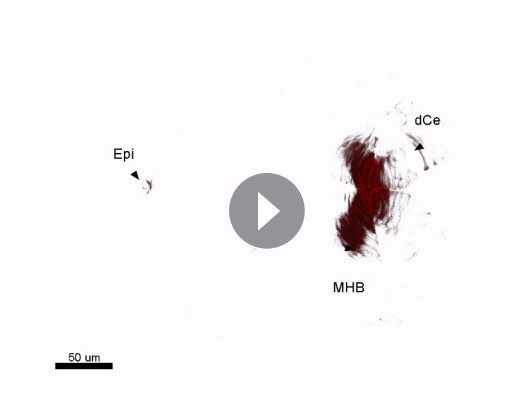

**Video 3.** Source (red) and distal target (green) regions of Wnt3 at 24 hpf.
https://elifesciences.org/articles/59489#video3

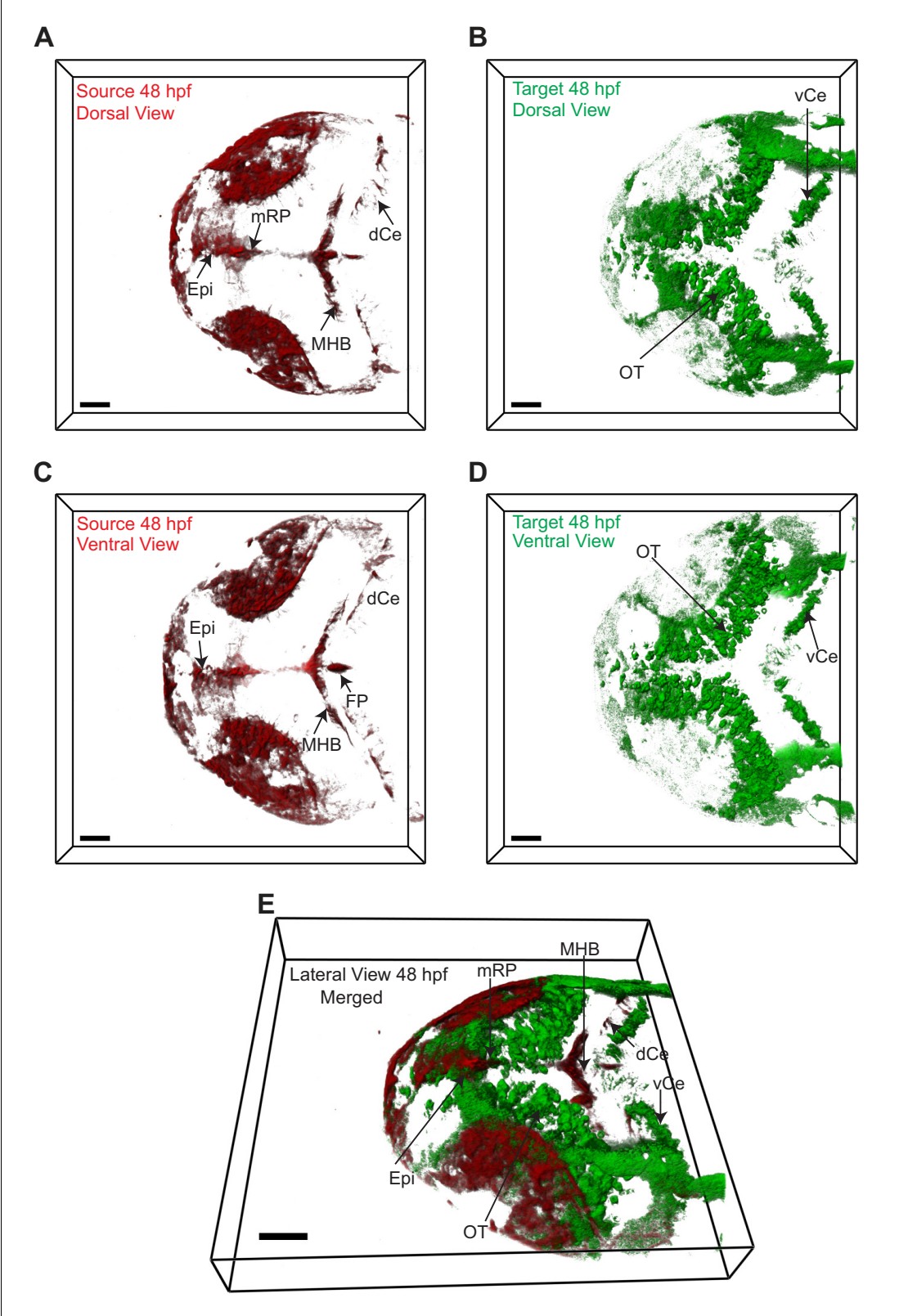

**Figure 3.** Wnt3 source and target regions at 48 hpf. 3D dorsal projection of Wnt3 (**A**) source regions at 48 hpf and (**B**) target regions at 48 hpf (top view). 3D ventral projection of Wnt3 (**C**) source regions at 48 hpf and (**D**) target regions at 48 hpf (bottom view). (**E**) 3D projection of Wnt3 source and target regions at 48 hpf (lateral view). See *Video 4* for a detailed view. dCe, dorsal regions of cerebellum; Epi, epithalamus; FP, floor plate; MHB,

*Figure 3 continued on next page*

*Figure 3 continued*

midbrain–hindbrain boundary; mRP, midbrain roof plate; OT, optic tectum; vCe, ventral regions of cerebellum. Images orientation: anterior to the left. Scale bar 40 μm.

The online version of this article includes the following figure supplement(s) for figure 3:

**Figure supplement 1.** Fluorescence correlation spectroscopy (FCS) measurements of Wnt3EGFP and PMTmApple in the Wnt3 target regions at 48 hpf.
**Figure supplement 2.** Expression of EGFP in Tg(−4.0*wnt3*:EGFP) at 48 hpf.
**Figure supplement 3.** Expression of *wnt3* transcripts and downstream wnt signaling transcription factors at 48 hpf.

analysis on mammalian cell lines (*Dijksterhuis et al., 2015*). The dynamics and conformation of proteins might differ significantly in vivo (*Lipinski and Hopkins, 2004*), and quantitative analysis of Wnt-Fzd interactions in live organisms is still lacking. Since in vitro genetic and biochemical assays reported that Wnt3 interacts strongly with Fzd1 (*Dijksterhuis et al., 2015*), we investigated the in vivo Wnt3EGFP-Fzd1mApple interaction and measured its binding affinity. For this purpose, we generated a transgenic line Tg(−4.0*wnt3*:Fzd1mApple) expressing Fzd1mApple, crossed it with the Wnt3EGFP expressing line, and studied in vivo interactions using quasi-PIE FCCS (*Figure 6A,B*). Quasi-PIE FCCS is an extension of FCCS, where the sample is simultaneously illuminated by a pulsed laser line and a continuous wave laser line of different wavelengths (*Padilla-Parra et al., 2011*; *Yavas et al., 2016*). This approach allows us to filter background, spectral cross-talk, and detector after pulsing while computing the auto- and cross-correlation functions (*Kapusta et al., 2012*). When quasi-PIE FCCS measurements were performed in embryos expressing Wnt3EGFP and Fzd1mApple, we obtained positive cross-correlations between the two channels, indicating the in vivo interaction of Wnt3EGFP with Fzd1mApple (*Figure 6C*). As a positive control, we used embryos expressing PMT-mApple-mEGFP, and as negative control, we used embryos expressing Wnt3EGFP and PMTmApple by crossing their respective transgenic lines (*Figure 6—figure supplement 1*). The auto- and cross-correlations were then fitted with *Equation (7)*, and the binding affinity was measured according to *Equation (12)* (see Materials and methods). We obtained an apparent dissociation constant ($K_d$) of 112 ± 15 nM indicating that Wnt3EGFP binds strongly with Fzd1mApple in vivo (*Figure 6D*). The measured in vivo $K_d$ for Wnt3-Fzd1 is comparable with the in vitro $K_d$ values reported for Wnts with Fzd1, which were in the range of 15–90 nM (*Dijksterhuis et al., 2015*).

Interestingly, Wnt3EGFP-Fzd1mApple interactions were only detected in the MHB and the dCe of the zebrafish brain at 48 hpf. No interactions were detected in the vCe or OT despite detecting Wnt3EGFP and Fzd1mApple in these regions (*Figure 6—figure supplement 2A*). Since the expression of the co-receptor *lrp5* corresponds to the specific regions where we detected interactions (*Willems et al., 2015*), we hypothesized that Lrp5 is necessary for the in vivo binding of Wnt3 to Fzd1. To test this, we knocked down the expression of *lrp5* using morpholinos (Mo) and checked for Wnt3EGFP-Fzd1mApple interactions in the MHB and dCe. We used *lrp5*MoUp, which targets the Exon2-Intron2 splice junction, and *lrp5*MoDown that targets Intron2-Exon3 splice junctions to knockdown *lrp5*. As a control, we used a mismatch morpholino containing five nucleotide substitutions (see Materials and methods for details). The efficacy of morpholinos was previously characterized in *Willems et al., 2015*. Upon knockdown of *lrp5,* we observed defective brain development (*Figure 6—figure supplement 2B*) and we did not detect any cross-correlations in MHB and dCe (*Figure 6E*). However, cross-correlations were obtained in the corresponding regions for untreated embryos and embryos treated with mismatch morpholino (*Figure 6—figure supplement 2C,D*). When we performed FRAP for Wnt3EGFP after *lrp5* knockdown, we obtained a faster recovery of ~1.5 min for Wnt3EGFP in the photobleached

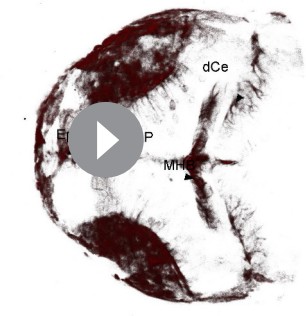

**Video 4.** Source (red) and distal target (green) regions of Wnt3 at 48 hpf.
https://elifesciences.org/articles/59489#video4

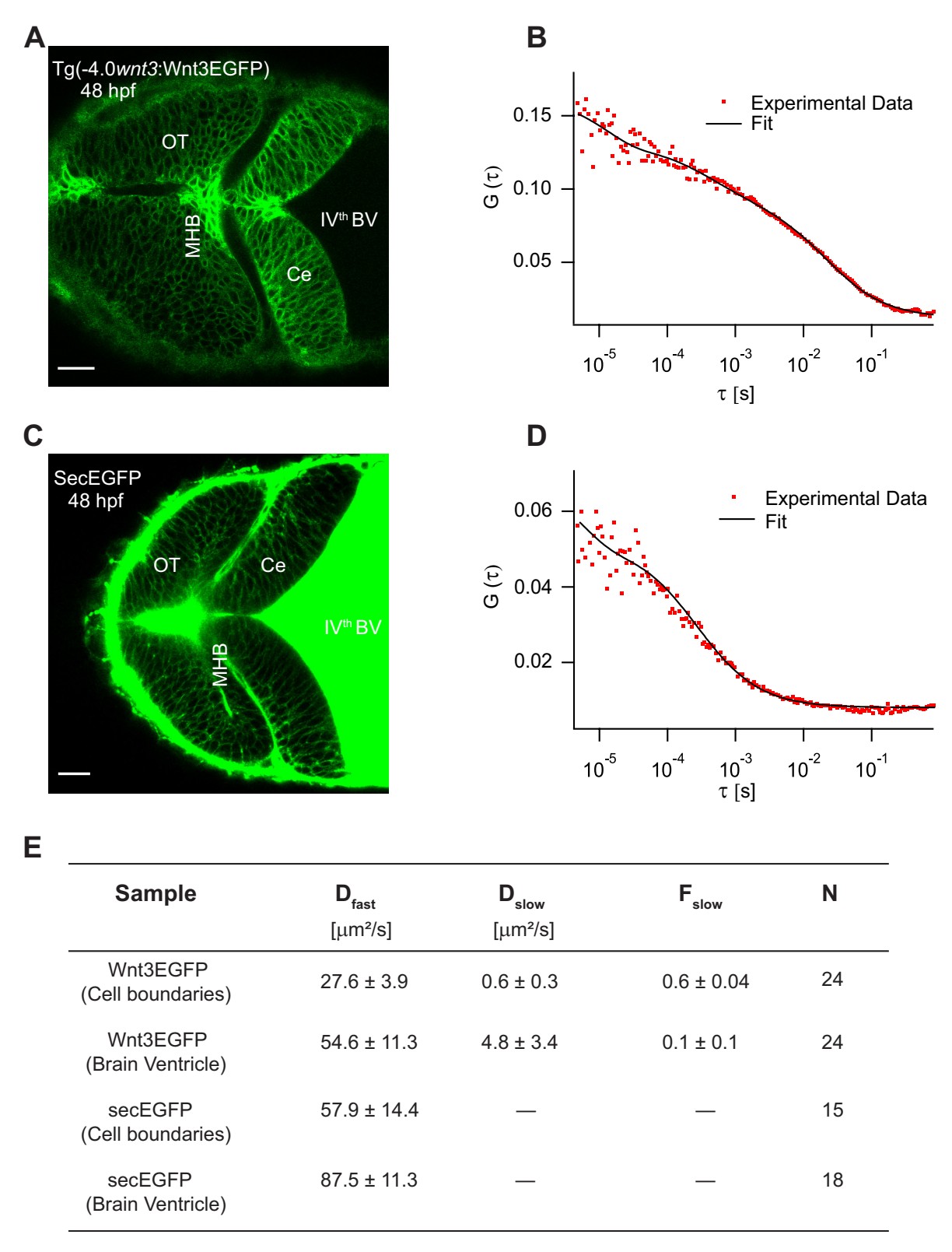

**Figure 4.** Characterizing the dynamics of Wnt3EGFP and secEGFP using fluorescence correlation spectroscopy (FCS). (**A**) Expression of Wnt3EGFP in Tg(−4.0*wnt3*:Wnt3EGFP) at 48 hpf. (**B**) Representative autocorrelation function (ACF; dots) and fitting (line) of a Wnt3EGFP measurement at a cell boundary. (**C**) Expression of secEGFP in the zebrafish brain at 48 hpf. (**D**) Representative ACF (dots) and fitting (line) of a secEGFP measurement at a cell boundary. (**E**) Table showing diffusion coefficients of the fast component ($D_{fast}$), slow component ($D_{slow}$) and the fraction of slow component ($F_{slow}$)
*Figure 4 continued on next page*

*Figure 4 continued*

for Wnt3EGFP and secEGFP measured by FCS. Measurements were performed in the cell borders of Ce, MHB, and OT; and in the BV. Data are mean ± SD; N = No of measurements. BV, brain ventricle; Ce, cerebellum; MHB, midbrain–hindbrain boundary; OT, optic tectum. Images orientation: anterior to left. Scale bar 30 μm.

The online version of this article includes the following source data and figure supplement(s) for figure 4:

**Source data 1.** Diffusion coefficients of Wnt3EGFP and secEGFP at cell borders and in the brain ventricle.

**Figure supplement 1.** Expression and dynamics of secEGFP under 4 kb *wnt3* promoter.

region with a $D_{global}$ of 2.8 ± 0.8 μm²/s, indicating that receptor binding slows Wnt3EGFP diffusion by a factor ~5–6 (*Figure 6—figure supplement 3*). These results suggest that the co-receptor Lrp5 is essential for in vivo Wnt3EGFP-Fzd1mApple interaction and that this interaction influences Wnt3EGFP diffusion.

## Discussion

Symmetry breaking and the development of an embryo into an organism require a finely balanced but robust position-sensitive control of cell behavior and differentiation. This is achieved by signaling molecules that are expressed in well-defined source regions and distribute to target tissues where they are recognized by their cognate receptors. Wnts are a class of molecules that fulfill this function and are involved in cell division, cell migration, apoptosis, embryonic axis induction, cell fate determination, and maintenance of stem cell pluripotency (*Clevers and Nusse, 2012*; *Logan and Nusse, 2004*). Misregulation of this process leads to developmental defects and diseases, including cancer. In this work, we investigated the in vivo action mechanism of Wnt3, a member of this family that is involved in the proliferation and differentiation of neural cells, with particular attention to the differentiation of source and target regions, the mode of transport, and the recognition of Wnt3 by its target receptor.

First, we analyzed the colocalization of Wnt3EGFP and PMTmApple expression in the double transgenic [Tg(−4.0*wnt3*:Wnt3EGFP) × Tg(−4.0*wnt3*:PMTmApple)] to map Wnt3 source and distal target regions at 24 hpf and 48 hpf. We categorized the MHB, Epi, and dCe as the source regions for Wnt3 at 24 hpf. We observed that the source regions expanded with time to include the mRP, FP, and dorso-lateral OT as source at 48 hpf. Interestingly, earlier studies had documented these

**Table 1.** Influence of heparan sulfate proteoglycans on the dynamics of Wnt3EGFP, LynEGFP, and secretedEGFP at 48 hpf. Data are mean ± SD.

| Sample | $D_{fast}$ (μm²/s) | $D_{slow}$ (μm²/s) | $F_{slow}$ | No. of measurements |
|---|---|---|---|---|
| Wnt3EGFP | 24.7 ± 4.8 | 0.6 ± 0.3 | 0.6 ± 0.02 | 47 |
| Wnt3EGFP + heparinase | 43.4 ± 7.6 | 0.4 ± 0.2 | 0.6 ± 0.04 | 63 |
| Wnt3EGFP + surfen | 42.0 ± 6.9 | 0.5 ± 0.3 | 0.6 ± 0.08 | 30 |
| LynEGFP | 39.1 ± 11.2 | 2.2 ± 0.6 | 0.7 ± 0.04 | 29 |
| LynEGFP + heparinase | 40.1 ± 9.5 | 2.7 ± 0.7 | 0.6 ± 0.04 | 35 |
| LynEGFP + surfen | 41.4 ± 6.9 | 3.1 ± 0.9 | 0.7 ± 0.06 | 18 |
| SecEGFP | 59.4 ± 9.4 | - | - | 30 |
| SecEGFP + heparinase | 56.9 ± 9.7 | - | - | 30 |
| SecEGFP + surfen | 53.5 ± 8.5 | - | - | 18 |

The online version of this article includes the following source data for Table 1:

**Source data 1.** Tg(−4.0*wnt3*:Wnt3EGFP) embryos treated by heparinase and surfen.(A) The expression of Wnt3EGFP after heparinase treatment. (B) Distribution of Dextran-TRITC coinjected with heparinase in the BV of Tg(−4.0*wnt3*:Wnt3EGFP) embryo. (C) Expression of Wnt3EGFP after surfen treatment. BV, brain ventricle; Ce, cerebellum. Images orientation: anterior to the left. Scale bar 50 μm.

**Source data 2.** Individual fluorescence correlation spectroscopy measurements of Wnt3EGFP, LynEGFP, and secEGFP embryos before and after heparan sulfate proteoglycan disruption.

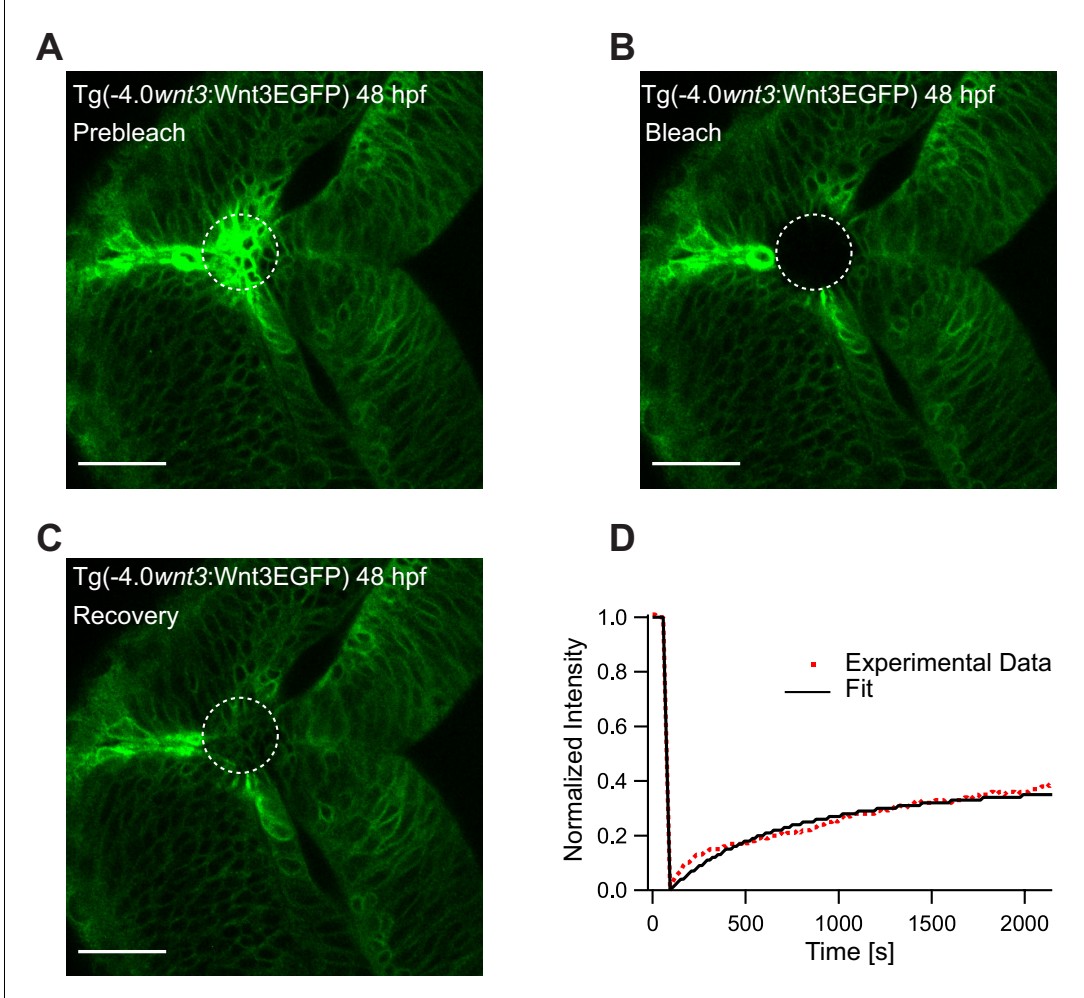

**Figure 5.** Representative fluorescence recovery of Wnt3EGFP at 48 hpf after photobleaching. (**A**) Expression of Wnt3EGFP in Tg(−4.0*wnt3*:Wnt3EGFP) at 48 hpf before photobleaching. (**B**) Photobleached region of Wnt3EGFP. (**C**) Recovery of fluorescence intensity in the bleached region due to diffusion of molecules from the neighboring unbleached regions. (**D**) Fluorescence recovery curve for Wnt3EGFP with a time constant ($\tau_{fast}$) of ~5 min and a mobile component fraction ($F_m$) of ~0.35. The average apparent global diffusion coefficient ($D_{global}$) measured for Wnt3EGFP was 0.5 ± 0.2 μm$^2$/s (N = 11). Fluorescence recovery after photobleaching for Wnt3EGFP at a distal target site showed similar recovery dynamics (*Figure 5—figure supplement 1*) whereas recovery after heparan sulfate proteoglycan disruption showed faster recovery (*Figure 5—figure supplement 2*). Orientation: anterior to the left. Scale bar 30 μm.

The online version of this article includes the following source data and figure supplement(s) for figure 5:

**Source data 1.** Individual fluorescence recovery after photobleaching measurements of Wnt3EGFP and secEGFP embryos.
**Figure supplement 1.** Representative fluorescence recovery of Wnt3EGFP at a distal target site after photobleaching.
**Figure supplement 2.** Representative fluorescence recovery of Wnt3EGFP at a distal target site after heparan sulfate proteoglycan (HSPG) disruption.
**Figure supplement 3.** Representative fluorescence recovery of secEGFP after photobleaching at 48 hpf.
**Figure supplement 4.** Representative fluorescence recovery of PMTmApple after photobleaching at 48 hpf.

regions as the primary signaling centers that control the development of the central nervous system (CNS). The brain midline, comprising of the roof plate and FP, represent the signaling glia that acts as the source of several secreted signals involved in neuronal specification (*Chizhikov et al., 2006*; *Jessell, 2000*; *Kondrychyn et al., 2013*). *Chizhikov and Millen, 2005* provided a comprehensive overview on how the roof plate governs the specification of the hindbrain, diencephalon, telencephalon, and spinal cord by producing BMP and Wnt proteins. Similarly, the importance of the MHB (also known as the isthmic organizer) in the morphogenesis of the zebrafish brain is also well studied (*Gibbs et al., 2017*; *Raible and Brand, 2004*; *Wurst and Bally-Cuif, 2001*). Our results, at a

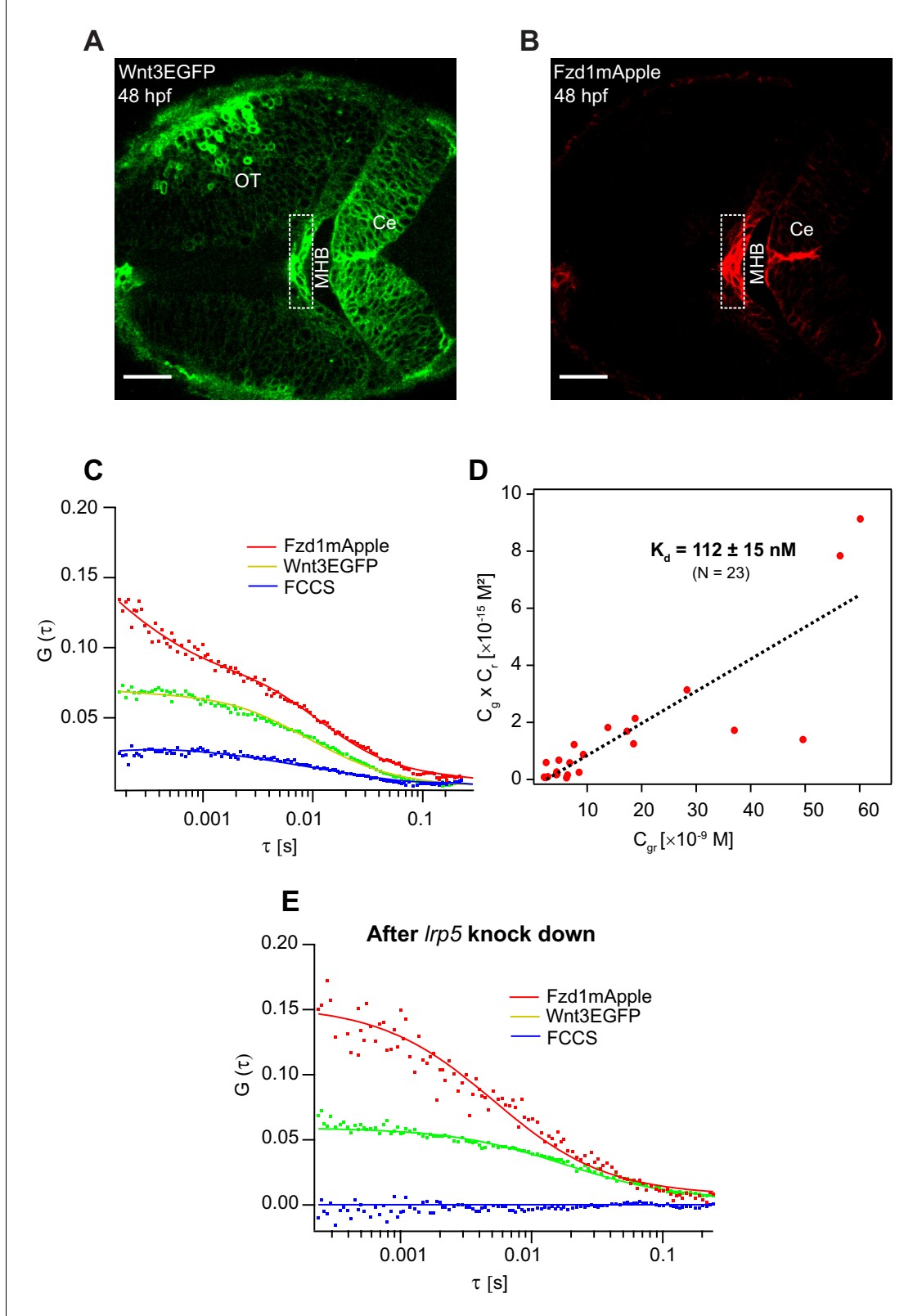

**Figure 6.** Investigation of in vivo Wnt3-Fzd1 binding by FCCS. Expression of (**A**) Wnt3EGFP and (**B**) Fzd1mApple in the double transgenic [Tg(−4.0*wnt3*: Wnt3EGFP)×Tg (−4.0*wnt3*:Fzd1mApple)] (anterior to the left). (**C**) Representative auto- and cross-correlation functions (dots) and fittings (lines) of a Wnt3EGFP-Fzd1mApple measurement at the indicated region. The cross-correlation function indicates Wnt3EGFP interacts with Fzd1mApple in vivo. (**D**) Determination of apparent dissociation constant (K$_d$) for Wnt3-Fzd1 interaction in vivo. $C_g$, $C_r$, and $C_{gr}$ represents the concentration of unbound
*Figure 6 continued on next page*

*Figure 6 continued*

Wnt3EGFP, unbound Fzd1mApple, and bound Wnt3-Fzd1 molecules respectively. The estimated apparent $K_d$ [$K_d = (C_g \times C_r)/C_{gr}$] for Wnt3-Fzd1 in vivo is 112 ± 15 nM (N = 23; $R^2$ = 0.85). (E) Representative auto- and cross-correlation functions (dots) and fittings (lines) of a Wnt3EGFP-Fzd1mApple measurement after knocking down *lrp5*. No cross-correlation indicates Wnt3-Fzd1 interaction is abolished after knockdown of *lrp5*. Scale bars 30 μm.

The online version of this article includes the following source data and figure supplement(s) for figure 6:

**Source data 1.** Apparent dissociation constant ($K_d$) calculation for Wnt3-Fzd1 interaction and fluorescence recovery after photobleaching recovery rates after *lrp5* knockdown.

**Figure supplement 1.** Representative FCCS measurements for positive and negative control in zebrafish.

**Figure supplement 2.** Role of Lrp5 coreceptor in Wnt3-Fzd1 binding.

**Figure supplement 3.** Representative fluorescence recovery of Wnt3EGFP in *lrp5* knocked-down embryos at 48 hpf after photobleaching.

---

molecular level, corroborate these functional studies, which examine the role of these signaling centers in coordinating brain development by producing critical signaling molecules.

As we identified the Wnt3 source regions based on the colocalization analysis of Wnt3EGFP and PMTmApple fluorescence signal, any weak mApple signal that is undetected by confocal imaging might underestimate the full range of Wnt3 source regions. Hence, we used FCS, a single molecule sensitive technique, to check if any weak PMTmApple intensity fluctuations are detected outside the classified source regions. When we performed FCS in the double transgenic [Tg(−4.0*wnt3*: Wnt3EGFP) × Tg(−4.0*wnt3*:PMTmApple)] embryos, we obtained autocorrelation functions for Wnt3EGFP but not for PMTmApple in the OT and ventral regions (*Figure 2—figure supplement 1* and *Figure 3—figure supplement 1*). Furthermore, the expression pattern of PMTmApple closely resembles the expression of Tg(−4.0*wnt3*:EGFP) (*Figure 2—figure supplement 2* and *Figure 3— figure supplement 2*), a reporter line driven by the 4 kb *wnt3* promoter expressing cytosolic EGFP in domains of endogenous *wnt3* transcripts (*Clements et al., 2009*; *Duncan et al., 2015*; *Teh et al., 2015*; *Figure 2—figure supplement 3*, *Figure 3—figure supplement 3*). The expression of *egfp* transcripts also closely mirrors the expression of *wnt3* transcripts, implying that the 4 kb *wnt3* promoter contains most, if not all, regulatory elements (*Teh et al., 2015*). While these results suggest that Tg(−4.0*wnt3*:PMTmApple) is indeed a faithful reporter of Wnt3 expression, we cannot exclude the possibility of underestimating the extent of Wnt3 source regions.

Our approach based on the analysis of the distribution of proteins in vivo enabled us to validate not only the source regions but also the obtained information regarding the distribution of signaling proteins in live samples. We identified the vCe and OT as the target regions to where Wnt3EGFP is transported. These identified target regions correspond to the 7 T-cell factor (TCF) responsive elements-driven nuclear localization sequence mCherry (NLS-mCherry) expression in the wnt reporter line Tg(*7xTcf-Xla.Siam*:NLS-mCherry) that is activated by endogenous Wnt ligands (*Moro et al., 2012*; *Figure 2—figure supplement 3* and *Figure 3—figure supplement 3*). However, the whole list of Wnt3 target sites could be longer. Recently, it was shown that Wnt5A transported in the cerebrospinal fluid regulates the development of the hindbrain (*Kaiser et al., 2019*). Since we also detect Wnt3EGFP diffusing in the BV (*Teh et al., 2015*), further investigation is required to detect additional less obvious target sites. Nevertheless, the characterization of Wnt3 source and target regions of this work clearly indicates the presence of discrete Wnt3-producing and -receiving cells in the developing brain of zebrafish embryos.

Second, we investigated the transport mechanism of Wnt3EGFP in the zebrafish brain by measuring its local and global diffusion coefficients using FCS and FRAP respectively. The transport mechanism not only influences signaling and function, but is of particular interest for Wnts as it is not clear how they can distribute over long distances despite their hydrophobic nature. Using FCS, we first quantified the in vivo diffusion of Wnt3EGFP along the cell boundaries and in the BV. In the BV, we found a fast diffusing component with $D_{fast}$ of 54.6 ± 11.3 μm$^2$/s and a slow component with $D_{slow}$ of 4.8 ± 3.4 μm$^2$/s (*Figure 4E*). The first component is similar to secEGFP and consistent with freely diffusing Wnt3EGFP, or at best Wnt3EGFP in a small complex, e.g. with a shuttling protein that hides the hydrophobic Wnt3 moiety and prevents its aggregation. The second component is much slower and hints at Wnt3EGFP associated with larger protein or lipid complexes and would be consistent with either exosomes or protein transport complexes. It will be interesting to address the exact nature of the aggregation and/or complexation state of Wnt3 in future studies. At the cell boundaries, we found two diffusive components for all Wnt3EGFP measurements: a slow component that

is consistent with membrane diffusion ($D_{slow}$ = 0.6 ± 0.3 μm²/s) and a fast component ($D_{fast}$ = 27.6 ± 3.9 μm²/s) much closer to the diffusion coefficient seen for secEGFP ($D$ = 57.9 ± 14.4 μm²/s). Due to resolution limitations of FCS, we could not unambiguously attribute this component to secreted Wnt3EGFP, as cytosolic Wnt3EGFP could also contribute to the fast diffusing component. Since Wnts has been shown to interact with HSPG (*Fuerer et al., 2010*; *Kirkpatrick and Selleck, 2007*; *Mii et al., 2017*), we disrupted HSPG by heparinase and surfen treatment, which should influence the diffusion of only extracellular Wnt3EGFP but not the putative cytosolic component. In subsequent measurements, $D_{fast}$ for Wnt3EGFP, but not for secEGFP and LynEGFP, increased almost twofold upon HSPG perturbation indicating that Wnt3EGFP indeed spreads by extracellular diffusion. If Wnt3EGFP (~73 kDa) diffuses freely similar to secEGFP (~33 kDa), one would expect a 30% difference in their diffusion coefficient ($D$ is inversely proportional to the cube root of the mass). As Wnt3EGFP interacts with HSPG it is slowed down in its movement. Thus it diffuses a factor ~2 (=57/28) slower than secEGFP. However, after HSPG disruption, secEGFP and Wnt3EGFP differ by a factor of 60/46 = 1.3 as expected. This implies that the diffusion coefficient for Wnt3EGFP increases as it is no longer retarded by HSPG, and not due to clearing up of free space. Recently, consistent with our experiments, *McGough et al., 2020* demonstrated how glypicans, a major family of HSPG, enable the spreading of Wingless in Drosophila by shielding the Wnt lipid moiety. Overall, these results are in line with Wnt transport models that support the extracellular distribution by HSPG (*Baeg et al., 2001*; *Han et al., 2005*; *Mii et al., 2017*; *Yan and Lin, 2009*). Similar results were also reported for Fgf8, which establishes a concentration gradient in zebrafish embryos by extracellular diffusion (*Yu et al., 2009*).

FRAP experiments conducted at multiple cell diameters removed from the source region corroborate these results. Fluorescence recovery took place in 5–8 min for Wnt3EGFPindicating transport over long distances. However, the estimated apparent global diffusion coefficient of Wnt3EGFP ($D_{global}$) was only 0.5 ± 0.2 μm²/s, a factor ~40–100 lower than the local diffusion coefficient in the interstitial spaces measured by FCS (27.6 ± 3.9 μm²/s) (*Figure 5*). This is in stark contrast to secEGFP $D_{global}$ which was estimated to be 13 ± 4 μm²/s and was reduced by only about a factor ~3–5 compared to FCS measurements of the same molecule with $D_{local}$ of 57.9 ± 14.4 μm²/s (*Figure 5—figure supplement 1*). As secEGFP is only a secreted control that diffuses in the same environment as Wnt3EGFP, it does not interact with HSPG or receptors, and the smaller reduction in the global versus the local diffusion coefficient for secEGFP is likely an effect of tortuosity (*Müller et al., 2013*). However, the much larger reduction of the global diffusion coefficient for Wnt3EGFP calls for a different explanation, possibly including transient binding to its receptors and HSPG (*Müller et al., 2013*). Subsequent experiments showed that HSPG disruption by heparinase or surfen increased Wnt3EGFP diffusion by a factor ~2 (*Table 1*, *Figure 5—figure supplement 2*), and *lrp5* knockdown increased the Wnt3EGFP $D_{global}$ by a factor ~5–6 (*Figure 6—figure supplement 3*). Overall this accounts for a reduction in global Wnt3EGFP diffusion by at least a factor 30–60, consistent with the 40–100-fold reduction seen by the comparison of short-range (FCS) and long-range (FRAP) diffusion of Wnt3EGFP in native conditions. Hence, our FCS and FRAP results are consistent and collectively implicate the extracellular diffusion of Wnt3EGFP mediated by HSPG and receptor binding to accomplish long-range dispersal in the developing zebrafish brain.

However, we cannot discount the possibility that Wnt3 might additionally assume other modes of spreading. It is possible that carrier proteins or exosomes also shuttle Wnt3 in the zebrafish brain as would be consistent with the second slow component of Wnt3EGFP diffusion found in the BV. Moreover, HSPG may also assist in the transfer of Wnt bearing exosomes or lipoproteins by acting as their binding sites. A study demonstrated how HSPG guides the clearance of very low-density lipoprotein (VLDL) by forming a complex with Lrp (*Wilsie and Orlando, 2003*). Correspondingly, Eugster et al. explained how the interaction of the Drosophila lipoprotein with HSPG might influence the long-range signaling of Hedgehog in Drosophila (*Eugster et al., 2007*). On the same note, it was also determined how the functional activity of exosomes and vesicles is dependent on HSPG (*Christianson and Belting, 2014*). Further examination is required to confirm if HSPG aids the transport of Wnt3 packaged in exosomes or lipoprotein particles in the zebrafish brain. Nevertheless, our findings illustrate how HSPG moderates the long-range extracellular spreading, and by extension the function, of Wnt3 in the zebrafish brain.

It is also important to note that any modifications with fluorescent proteins are liable to have an influence on Wnt3 behavior. While we have previously shown that Wnt3EGFP is still biologically

functional (*Teh et al., 2015*), EGFP could also change protein diffusion or transport. The relationship between the diffusion coefficient (D) of a molecule and its molecular weight (M) is :$D \propto 1\sqrt[3]{M}$. Possible shape effects could potentially slow down the molecule even more. The mass of EGFP is ~27 kDa, and the mass of Wnt3 ~40 kDa. Hence, the measured diffusion coefficient of Wnt3EGFP is expected to be ~15% slower than the diffusion coefficient of Wnt3 molecule without the EGFP tag. However, this difference is relatively small and typically within the margins of the standard deviation of our measurements. As the standard deviation of our measurements includes measurement errors as well as the heterogeneity of the sample, we do not expect the label to have a strong influence. This is seen when comparing the diffusion of secEGFP and Wnt3EGFP, which show a difference of 30% in the diffusion coefficient as expected from their mass. Finally, as already discussed only Wnt3EGFP shows HSPG-dependent retardation in diffusion and Fzd1 binding. Therefore, we think that the influence on Wnt3 diffusion by the tag is within the normal variation of diffusion due to sample heterogeneity and is not influencing its interaction and function strongly.

Once Wnt ligands reach the target cells, the next question is how they interact with their target receptors. As we had established that it is highly unlikely for Wnts to diffuse in the interstitial spaces freely, they must be released from their chaperones or HSPG in order to interact with their receptors. One possible hand-off mechanism is the competitive binding of Wnts to their target receptors with a higher binding affinity (*Naschberger et al., 2017*; *Wilson, 2017*). Furthermore, the binding affinity of the Wnt-receptor complex also modulates their range and magnitude in vivo. Hence, we measured the in vivo binding affinity for Wnt3 with a potential target receptor, Fzd1 using quasi-PIE FCCS. We obtained an apparent $K_d$ of 112 ± 15 nM in vivo, implying a strong interaction. However, the actual $K_d$ might be even lower as the concentration of the endogenous proteins, and the photophysics of the fluorophore affects the estimated $K_d$ (*Foo et al., 2012*). Nonetheless, it is an estimate of the native in vivo Wnt3-Fzd1 binding in their physiological condition, which is consistent with the results of in vitro experiments (*Dijksterhuis et al., 2015*). Interestingly, we also noticed that the interaction of Wnt3 with Fzd1 was dependent on the expression of the co-receptor Lrp5. We did not detect any cross-correlations when the expression of *lrp5* was knocked down and the $D_{global}$ for Wnt3EGFP increased by a factor ~3–5. From this result, it appears that Lrp5 is an essential component in facilitating the interaction of Wnt3 with Fzd1 with a significant influence on the diffusion coefficient and the long-range spreading of Wnt3. Hence, it is of interest to measure the $K_d$ for Wnt3-LRP5 in the future and verify if the co-receptor is involved in the hand-off of Wnt from the carrier proteins and HSPG to its receptor. Note that Fzd1mApple expression in our transgenic line was driven by a 4 kb *wnt3* promoter that mimicked the regular expression of Wnt3. While useful methodologically to measure auto- and juxtacrine interactions of Wnt3-Fzd1, additional work is needed in measuring the in vivo binding affinities for Wnt3 with Fzd receptors expressed under the control of their native promoters.

In conclusion, our results show the presence of distinct Wnt3 source and target regions in the developing zebrafish brain, and that Wnt3 is distributed from its source to target by extracellular diffusion. We observed that the diffusion of Wnt3 is retarded by a factor 3–5 due to tortuosity, a factor 5–6 due to receptor binding, and a factor ~2 due to HSPG, thus leading to a total reduction of a factor 30–60 when comparing Wnt3EGFP short-range (~28 μm²/s as measured by FCS) to long-range diffusion (~0.5 μm²/s. as measured by FRAP). This reconciles the diffusion coefficients derived from FCS and FRAP and indicates that the major part if not all the reduction seen for long-range compared to short-range diffusion of Wnt3 is explainable by tortuosity, receptor binding, and interactions with HSPG present in the interstitial spaces.

Finally, we demonstrated that the co-receptor Lrp5 drives the in vivo interaction of Wnt3 with Fzd1, and quantitatively determined their affinity. This demonstrates that the presence of proteins alone, be it signaling molecules or receptors, as determined by fluorescence microscopy does not report on the actual signaling but it is necessary to measure interactions or downstream signaling to differentiate the concentration from the functional distribution of signaling molecules. Overall, our findings provide a general outline of Wnt3 signaling in the zebrafish brain from expression and transport to target binding, which set a starting point for the quantitative investigation of the Wnt3 interaction network during zebrafish brain development.

# Materials and methods

## Key resources table

| Reagent type (species) or resource | Designation | Source or reference | Identifiers | Additional information |
|---|---|---|---|---|
| Gene (*Danio rerio*) | fzd1 | Ensembl Zebrafish (GRCZ11) | ENSDART00000179533.2 | Transcript: fzd1-201 |
| Genetic reagent (*Danio rerio*) | Tg(−4.0wnt3:Wnt3EGFP) | *Teh et al., 2015* https://doi.org/10.1242/dev.127589 | ZDB-TGCONSTRCT-150922–7 | The transgenic zebrafish line expresses functional Wnt3EGFP driven by the 4 kb *wnt3* promoter |
| Genetic reagent (*Danio rerio*) | Tg(−4.0*wnt3*:EGFP) | *Teh et al., 2015* https://doi.org/10.1242/dev.127589 | ZDB-TGCONSTRCT-150922–8 | The transgenic zebrafish line expresses cytosolic EGFP driven by the 4 kb *wnt3* promoter |
| Genetic reagent (*Danio rerio*) | Tg(−4.0*wnt3*:PLMTmApple) [referred as Tg(−4.0*wnt3*:PMTmApple) in this paper] | This paper | ZDB-TGCONSTRCT-201104-1 | Plasma membrane targeting sequence tagged with mApple (PMTmApple) in this transgenic line marks the plasma membrane of the Wnt3-producing cells<br><br>(refer to "Generation of transgenic lines and zebrafish maintenance" for additional details). |
| Genetic reagent (*Danio rerio*) | Tg(−4.0*wnt3*:Fzd1mApple) | This paper | ZDB-TGCONSTRCT-201104–2 | The 4 kb *wnt3* promoter drives Fzd1mApple expression in Wnt3-producing cells<br><br>(refer to "Generation of transgenic lines and zebrafish maintenance" for additional details). |
| Genetic reagent (*Danio rerio*) | Tg(−8.0*cldnB*:lynEGFP) | *Haas and Gilmour, 2006* 2 https://doi.org/10.1016/j.devcel.2006.02.019 | ZDB-TGCONSTRCT-070117–15 | |
| Genetic reagent (*Danio rerio*) | Tg(*7xTcf-Xla.Siam:NLS-mCherry*) | *Moro et al., 2012* | ZDB-TGCONSTRCT-110113–2 | This construct uses seven TCF responsive elements fused to the *Xenopus laevis* siamois minimal promoter to drive expression of NLS-mCherry |
| Recombinant DNA reagent | pminiTol2-4kbwnt3 pro-PMTmApple | This paper | | This plasmid is used to generate Tg(−4.0wnt3: PMTmApple) transgenic zebrafish<br><br>(refer to "Generation of transgenic lines and zebrafish maintenance" for additional details). |
| Recombinant DNA reagent | pminiTol2-4kb wnt3 pro-Fzd1mApple | This paper | | This plasmid is used to generate Tg(−4.0wnt3: Fzd1mApple) transgenic zebrafish<br><br>(refer to "Generation of transgenic lines and zebrafish maintenance" for additional details). |

*Continued on next page*

*Continued*

| Reagent type (species) or resource | Designation | Source or reference | Identifiers | Additional information |
|---|---|---|---|---|
| Recombinant DNA reagent | pminiTol2-4kbwnt 3pro-secEGFP | This paper | | This plasmid together with Tol2 transposase mRNA is microinjected into 1–2 cell stage zebrafish embryo for somatic expression of secGFP in Wnt3-positive domains<br><br>(refer to "Generation of transgenic lines and zebrafish maintenance" for additional details). |
| Sequence-based reagent | lrp5MoUp (Gene Tools) | *Willems et al., 2015* https://doi.org/10.1371/journal.pone.0131768 | | AGCTGCTCTTACAGTTTGTAGAGAG (25) |
| Sequence-based reagent | lrp5MoDown (Gene Tools) | | | CCTCCTTCATAGCTGCAAAAACAAG (25) |
| Sequence-based reagent | mmlrp5 (Gene Tools) | | | AGGTGCTGTTAGAGTTTCTAGACAG (25) |
| Chemical compound, drug | Heparinase I from *Flavobacterium heparinum* | Merck | Cat# H2519 | |
| Chemical compound, drug | Surfen Hydrate | Merck | Cat# S6951 | A heparan sulfate antagonist |
| Software, algorithm | Imaris | Oxford Instruments | RRID:SCR_007370 | Check Materials and methods subsection Colocalization analysis |

## Fluorescence correlation spectroscopy

The molecular movement of fluorescently labeled molecules will cause fluorescence fluctuations during their entry and exit in a small open observation volume. These fluctuations contain the information about the dynamics of these molecules. In confocal FCS the confocal volume of the microscope setup defines the observation volume. The measured intensity trace is autocorrelated to extract the average concentrations and diffusion coefficients of the molecule in the sample. The autocorrelation function (ACF), G (τ), is given by

$$G(\tau) = \frac{\langle F(t) \cdot F(t+\tau) \rangle}{\langle F(t) \rangle \cdot \langle F(t+\tau) \rangle} \tag{1}$$

where F(t) is the fluorescence intensity at time $t$, $\tau$ is the lag time, and $\langle \ldots \rangle$ represents the time average. For a Gaussian illumination profile, G($\tau$) for a three-dimensional free diffusion process with a single component and triplet state can be written as

$$G(\tau)_{3D,1p,1t} = \frac{1}{N} \left( 1 + \frac{\tau}{\tau_d} \right)^{-1} \cdot \left[ 1 + \frac{1}{K^2} \left( \frac{\tau}{\tau_d} \right) \right]^{-\frac{1}{2}} \cdot f_{trip}(\tau) + G_\infty \tag{2}$$

Here $N$ is the mean number of molecules in the observation volume and is inversely proportional to the amplitude of the ACF *G(0)*; $\tau_d$ is the diffusion time of the molecule; $G_\infty$ is the convergence at long lag times; $K$ is the structure factor that denotes the shape of the confocal volume

$$K = \frac{\omega_z}{\omega_{xy}} \text{ and } V_{eff} = \pi^{3/2} \omega_{xy}^2 \omega_z \tag{3}$$

where $\omega_z$ and $\omega_{xy}$ are the $1/e^2$ radii of the PSF in the axial and radial direction; and $f_{trip}$ ($\tau$) is the triplet function that accounts for the fraction of particles in the triplet state ($F_{trip}$) with a triplet relaxation time of $\tau_{trip}$, and it is represented as

$$f_{trip}(\tau) = 1 + \frac{F_{trip}}{1 - F_{trip}} e^{-\frac{\tau}{\tau_{trip}}} \tag{4}$$

If two diffusing components are present, then the correlation function for two component 3D diffusion process $G(\tau)_{3D,2p,1t}$ is

$$G(\tau)_{3D,2p,1t} = \frac{1}{N}\left\{(1-F_2)\left(1+\frac{\tau}{\tau_{d1}}\right)^{-1}\left[1+\frac{1}{K^2}\left(\frac{\tau}{\tau_{d1}}\right)\right]^{-\frac{1}{2}} + F_2\left(1+\frac{\tau}{\tau_{d2}}\right)^{-1}\left[1+\frac{1}{K^2}\left(\frac{\tau}{\tau_{d2}}\right)\right]^{-\frac{1}{2}}\right\}f_{trip}(\tau) + G_\infty \quad (5)$$

where $F_2$ is the fraction of the second component. For a 2D diffusion process such as on a membrane, the fitting *Equations (2) and (5)* become

$$G(\tau)_{2D,1p,1t} = \frac{1}{N}\cdot\left(1+\frac{\tau}{\tau_d}\right)^{-1}\cdot f_{trip}(\tau) + G_\infty \quad (6)$$

$$G(\tau)_{2D,2p,1t} = \frac{1}{N}\left\{(1-F_2)\left(1+\frac{\tau}{\tau_{d1}}\right)^{-1} + F_2\left(1+\frac{\tau}{\tau_{d2}}\right)^{-1}\right\}f_{trip}(\tau) + G_\infty \quad (7)$$

For FCS measurements, the system was first calibrated with Atto 488 dye for 488 nm and 485 nm laser lines and Atto 565 for 543 nm laser line. The known diffusion coefficient for the dye was 400 µm²/s at room temperature. The obtained correlation function was fit using *Equation (2)* and the free fit parameters were $N$, $\tau$, $\tau_{trip}$, $F_{trip}$, and $G_\infty$. The $K$ value and $V_{eff}$ were calculated using *Equation (3)*. The samples were dechorionated, anesthetized by Tricaine, and mounted in 1% low melt agarose in a No. 1.5 glass bottom MatTek petri dishes. The acquisition time for the measurements was 60 s and all measurements were performed at room temperature. For FCS measurements, we used fit models as determined by Bayes inference-based model selection (*Sun et al., 2015*). Bayes model selection aids in determining the most suitable model given the data and its noise. It corrects for highly correlated noise and model complexity and appropriately prevents overfitting. The most likely fit models for Wnt3EGFP, secEGFP, and LynEGFP in zebrafish embryos were determined in *Sun et al., 2015*. Accordingly, we used 2D-2particle-1triplet model (*Equation 7*) to fit for Wnt3EGFP, LynEGFP, and Fzd1mApple, and 2D-1particle-1triplet model (*Equation 2*) for secEGFP expressing embryos. The measurements in the BV were fit using 3D-2particle-1triplet model (*Equation 5*) for Wnt3EGFP and 3D-1particle-1triplet model (*Equation 2*) for secEGFP.

## Quasi PIE fluorescence cross-correlation spectroscopy

FCCS is a valuable tool to study biomolecular interactions in live samples. When two interacting molecules tagged with spectrally different fluorophores transit through the observation volume, the intensity fluctuations from the two channels can be cross-correlated to obtain the cross-correlation function $G_x(\tau)$ given by:

$$G_x(\tau) = \frac{\langle F_g(t)\cdot F_r(t+\tau)\rangle}{\langle F_g(t)\rangle\cdot\langle F_r(t)\rangle} \quad (8)$$

where $F_g$ and $F_r$ are the fluorescence intensity in the green and red channels, respectively.

For our FCCS measurements to detect Wnt3-Fzd1 interactions, we used an interleaved pulsed 485 nm laser line and a continuous wave 543 nm laser line to obtain the auto- and cross-correlation functions. This allowed us to apply statistical filtering (*Kapusta et al., 2012*) that helped in eliminating spectral cross-talk, background signal, and detector after-pulsing based on fluorescence lifetime correlation spectroscopy (FLCS) as detailed in *Padilla-Parra et al., 2011*. This is called quasi-PIE FCCS (*Yavas et al., 2016*).

Taking into account the background and spectral cross-talk, the amplitude of the ACF in the green channel $G_G(0)$, red channel $G_R(0)$, and the amplitude of the CCF $G_x(0)$ can be written as:

$$G_G(0) = \frac{\left(\eta_{g,G}\right)^2 C_g + \left(\eta_{r,G}\right)^2 C_r + \left(q_g\eta_{g,G} + q_r\eta_{r,G}\right)^2 C_{gr}}{N_A V_{eff}\left[\eta_{g,G}C_g + \eta_{r,G}C_r + \left(q_g\eta_{g,G} + q_r\eta_{r,G}\right)C_{gr} + \frac{\beta_G}{N_A V_{eff}}\right]^2} \quad (9)$$

$$G_R(0) = \frac{\left(\eta_{g,R}\right)^2 C_g + \left(\eta_{r,R}\right)^2 C_r + \left(q_g \eta_{g,R} + q_r \eta_{r,R}\right)^2 C_{gr}}{N_A V_{eff} \left[\eta_{g,R} C_g + \eta_{r,R} C_r + \left(q_g \eta_{g,G} + q_r \eta_{r,G}\right) C_{gr} + \frac{\beta_R}{N_A V_{eff}}\right]^2}$$

(10)

$$G_x(0) = \frac{\eta_{g,R}\eta_{g,G} C_g + \eta_{r,G}\eta_{r,R} C_r + \left(q_g \eta_{g,R} + q_r \eta_{r,R}\right)^2 C_{gr}}{N_A V_{eff}\left[\eta_{g,R} C_g + \eta_{r,R} C_r + \left(q_g \eta_{g,G} + q_r \eta_{r,G}\right) C_{gr} + \frac{\beta_R}{N_A V_{eff}}\right]^2} \left[\eta_{g,R} C_g + \eta_{r,R} C_r + \left(q_g \eta_{g,G} + q_r \eta_{r,G}\right) C_{gr} + \frac{\beta_R}{N_A V_{eff}}\right]$$

(11)

Here $\eta_{g,G}$ and $\eta_{r,R}$ represent the mean counts per particle per second (cps) for EGFP in the green and mApple in the red channels, respectively. For our samples we obtained a $\eta_{g,G}$ of ~1900 Hz and $\eta_{r,R}$ of ~1400 Hz. $\beta_G$ and $\beta_R$ represent the count rates of background collected in the green and red channels, respectively. $\beta_R$ measured from blank WT embryo was ~400 Hz while FLCS correction eliminated the background in the green channel ($\beta_G = 0$). $N_A$ is the Avogadro's number and $V_{eff}$ represents the effective confocal volume from calibration. $\eta_{r,G}$ and $\eta_{g,R}$ denote the cross-talk in the green and red channels, respectively, which was efficiently eliminated by quasi PIE FCCS ($\eta_{r,G}$ and $\eta_{g,R} = 0$). $q_g$ and $q_r$ are the correction factors due to FRET and quenching. Since the cps for Wnt3EFP and Fzd1mApple in their respective transgenics were same as that in double transgenic line, $q_g$ and $q_r$ were set to 1. *Equations 9–11* were solved for $C_g$, $C_r$, and $C_{gr}$, which denote the concentration of the free green, free red, and bound molecules in the observation volume, respectively. Using $C_g$, $C_r$, and $C_{gr}$ the dissociation constant ($K_d$) for the interaction which can be determined using *Equation 12*:

$$K_d = \frac{C_g \cdot C_r}{C_{gr}}$$

(12)

## Confocal microscope setup

An Olympus FV 1200 laser scanning confocal microscope (IX83; Olympus, Japan) integrated with a PicoQuant time resolved LSM upgrade kit (Microtime 200; GmbH, Germany) was used in this work. The sample was illuminated using a 488 nm laser beam (for EGFP) and 543 nm laser beam (for mApple) which was reflected to the back focal plane of an Olympus UPLSAPO 60X/1.2 NA water immersion objective. For all the experiments, the intensity of the laser before the objective was less than 20 µW. The emitted signal passes through a 120 µm pinhole before being filtered by an Olympus 510/23 emission filter (for EGFP) or an Olympus 605/55 emission filter (for mApple) and eventually directed to a PMT detector for imaging. For FCS measurements, 510/23 emission filter (Semrock, USA) and 615DF45 filter (Semrock, USA) were used, and the filtered emissions were recorded using a single photon sensitive avalanche photodiodes (SAPD) (SPCM-AQR-14; PerkinElmer). The recorded signal was then processed using SymPhoTime 64 (PicoQuant, Germany) to compute the autocorrelation function. For FCCS measurements, the sample was simultaneously illuminated with a pulsed 485 nm laser (LDH-D-C-488; PicoQuant) operated at 20 MHz repetition rate and a continuous 543 nm laser. The emission was separated using 560 DCLP dichroic mirror and directed to the 510/23 emission filter (Semrock, USA) and the 615DF45 filter (Semrock, USA). The signal recorded by SAPD was then analyzed by SymphoTime to generate the auto- and cross-correlations. The auto- and cross-correlation functions were fit with a custom written program on Igor Pro. Any brightness or contrast adjustments for the images were performed equally across all data.

## Colocalization analysis

For the colocalization analysis of Wnt3EGFP and PMTmApple, confocal z-stacks of step size 0.5 µm were obtained with identical acquisition settings. An automatic threshold algorithm detailed in *Zhu et al., 2016* was implemented to segment the data. The algorithm uses the correlation quotient to select an optimal threshold for segmentation as described in *Li et al., 2004*. Following the segmentation, the colocalization for each pixel was calculated based on ICA, the distance weight, and intensity weight (*Li et al., 2004*; *Zhu et al., 2016*). Finally, a pair of masks for the colocalized and non-colocalized pixels were generated. The colocalized pixels and non-colocalized pixels were used to construct 3D images of the source and target regions, respectively. The 3D images were built

using '3D View' module Imaris 9.5.0 (Oxford Instruments). The display setting was set to white background and the 3D reconstructed images are represented in 'Normal Shading' mode for improving contrast in *Figures 2* and *3* and *Videos 3* and *4*.

## Fluorescence recovery after photo bleaching

FRAP measurements were performed on an Olympus FV3000 laser scanning microscope. The mounted samples were imaged with a UPLSAPO 60X/1.2 NA water immersion objective (at 1.5× zoom) or a UPLSAPO 30X/1.05 NA silicone oil immersion objective (at 3× zoom). We used a 488 nm diode laser (for Wnt3EGFP and secEGFP) or a 561 nm diode laser (for PMTmApple) for our experiments. A DM 405/488/561/640 dichroic mirror separated the excitation and emission beams. The signal from the sample after passing through the dichroic mirror was filtered by a BP 510–550 emission band pass filter for the 488 nm laser beam, and by a BP 575–625 emission band pass filter for the 561 nm laser beam. The pinhole size was adjusted to 1 AU. For FRAP, five pre-bleach frames were obtained before irreversibly photo bleaching a circular region of interest (ROI) for 30 s. The fluorescence intensity recovery in the photobleached region was recorded for 30 min. The images were then analyzed using the FRAP module in the Olympus CellSens software. A reference region on the sample but outside the ROI was selected to correct for photo bleaching, and another reference region outside the sample was selected for background correction. The software then plotted an FRAP recovery curve for the ROI and fitted the FRAP curve with a double exponential fit to obtain the time constants for the fast ($\tau_{fast}$) and the slow ($\tau_{slow}$) component. The diffusion coefficient ($D_{global}$) was calculated using *Equation 13* (*Kang et al., 2015*; *Koppel et al., 1976*), as described in the PicoQuan Practical Manual for Fluorescence Microsopy Techniques. However, it must be noted that this is an apparent estimate of the $D_{global}$ as the distribution of fluorophores is not homogeneous, and we assume there is no diffusion during photo bleaching process.

$$D_{global} = \frac{r^2}{4\tau_{fast}} \tag{13}$$

## Generation of transgenic lines and zebrafish maintenance

To generate Tg(−4.0*wnt3*:PLMTmApple) [referred as Tg(−4.0*wnt3*:PMTmApple)] transgenic zebrafish, the 45 bp plasma membrane targeting-sequence (PMT) (ATGGGCTGCTTCTTCAGCAAGCGGCGGAAGGCCGACAAGGAGAGC) was cloned upstream and in-frame with mApple to generate PMTmApple open reading frame (ORF). The DNA fragment was subcloned into the 4-kbWnt3EGFP-miniTol2 recombinant plasmid (*Teh et al., 2015*) using Gibson assembly by replacing the Wnt3EGFP ORF with PMTmApple to give 4-kbPMT-mApple-miniTol2 recombinant plasmid.

To generate Tg(−4.0*wnt3*:Fzd1mApple), zebrafish *fzd1* ORF (1617 bp; ENSDARG00000106062) was amplified by RT-PCR and subcloned into pGemTeasy. The Fzd1mApple DNA fragment was constructed by removing the Fzd1 stop codon and inserting in-frame (GGGS)two linker sequence (GGAGGAGGATCAGGAGGAGGATCA) tagged with mApple to Fzd1 C terminal by Gibson assembly. This DNA fragment was then subcloned into the 4-kbWnt3EGFP-miniTol2 recombinant plasmid using Gibson assembly by replacing the Wnt3EGFP ORF with Fzd1mApple to give 4-kbFzd1mApple-miniTol2 recombinant plasmid.

Stable *wnt3* promoter-driven transgenic lines were generated as stated (*Balciunas et al., 2006*) by co-injection of transposase mRNA and 4-kbPMT-mApple-miniTol2 recombinant plasmid; co-injection of transposase mRNA and 4-kbFzd1mApple-miniTol2 recombinant plasmid, to generate Tg(−4.0*wnt3*:PMTmApple) and Tg(−4.0*wnt3*:Fzd1mApple) transgenic lines respectively.

For secEGFP measurements, embryos were injected with secEGFP mRNA (a gift from Prof. Karuna Sampath) at the 1–2 cell stage and subsequently screened for fluorescence. Additionally, secEGFP sequence was also subcloned into the 4-kbWnt3EGFP-miniTol2 recombinant plasmid by replacing Wnt3EGFP ORF with secEGFP to give 4-kbsecEGFP-miniTol2 recombinant plasmid which was co-injected with Tol2 transposase at 1–2 cell stage for 4 kb wnt3 promoter-driven somatic expression of secGFP. The embryos were subsequently screened before measurements.

Additional transgenic lines used are Tg(−8.0*cldn*B:lynEGFP) for in vivo imaging of membrane-tethered EGFP expression in the cerebellum (*Haas and Gilmour, 2006*) and Tg(−4.0*wnt3*:EGFP) for in vivo cytosolic EGFP expression in the domains of endogenous Wnt3 expression. Wnt3EGFP

expression in the brain was imaged using Tg(−4.0*wnt3*:Wnt3EGFP)F2 (*Teh et al., 2015*). The Tg (*7xTCF-Xla.Sia*:NLS-mCherry) embryos were a gift from Tom Carney's group.

Transgenic adult zebrafish and embryos were obtained from zebrafish facilities in the Institute of Molecular and Cell Biology (Singapore) and National University of Singapore. The Institutional Animal Care and Use Committee (IACUC) in Biological Resource Center (BRC), A*STAR, Singapore (IACUC #161105) and the National University of Singapore (IACUC# BR18-1023) have approved the entire study. Spawned transgenic embryos were staged as described (*Kimmel et al., 1995*). Embryos older than 30 hpf were treated with 1-phenyl-2-thiourea at 18 hpf to prevent formation of melanin.

### Morpholino injection

The injected dose of *lrp5* splice-blocking Morpholinos (MOs; Gene Tools, Corvalis, USA) lrp5MoUp (AGCTGCTCTTACAGTTTGTAGAGAG) targeting the Exon2-Intron2 splice junction and lrp5MoDown (CCTCCTTCATAGCTGCAAAAACAAG) targeting the Intron2-Exon3 splice junction were conducted in accordance to published research (*Willems et al., 2015*). As control, mismatch morpholino (mm*lrp5*) containing five nucleotide substitutions (AGgTGCTgTTAgAGTTTcTAGAcAG) was used (*Willems et al., 2015*).

### Heparinase injection into the zebrafish BV and surfen treatment

Heparinase I from *Flavobacterium heparinum* (Merck) was dissolved in PBS to 1 U/µl and stored as frozen aliquots. For microinjection into the BV, MS-222 (Merck) anesthetized 48hpf zebrafish embryos were laterally mounted in 1% low gelling agarose (Merck). Reaction mix containing 0.1 U/µl heparinase I and 70,000 MW Dextran-Tetramethylrhodamine (ThermoFisher Scientific) was injected into the fourth ventricle of immobilized embryo. Injected embryos were freed from agarose and allowed to recover in glass bottomed dishes prior to imaging.

For surfen treatment, 24 hpf embryos were incubated in 3 µM surfen hydrate (Merck) in 1% DMSO in accordance with *Naini et al., 2018* for 24 hr and measured at 48 hpf.

### Sample selection, quantification, and statistics

Healthy embryos at their corresponding developmental stages were selected and screened for fluorescence. The embryos were then dechorionated, treated with Tricaine (3-amino benzoic acidethylester, Sigma), and mounted on a 35 mm 1.5 coverglass bottom dishes (MaTek, USA) for imaging and measurements. For FCS measurements, a maximum of 10 measurements were performed on a single embryo to reduce phototoxicity effects. No statistical tests were performed to predetermine sample size. A minimum for 15 measurements were analyzed for all FCS experiments using a minimum of four different embryos. All data are represented as mean ± SD. For comparing the effects of heparinase and surfen treatment on the dynamics of Wnt3EGFP, LynEGFP, and secEGFP with untreated embryos, larger sample sizes were used for more accuracy. The heparinase/surfen-treated and -untreated embryos were maintained at identical conditions. Unpaired two-sided t-test was performed to find the statistical significance between the diffusion coefficients of untreated- versus heparinase/surfen-treated embryos and p-values were considered statistically significant at $p < 0.05$. All graphs were plotted and fitted on Igor Pro. For FRAP, only one measurement was performed on each embryo to reduce the phototoxicity effects. The embryos used for analysis in imaging, FCS, and FRAP measurements were from at least three different experiment sessions.

## Acknowledgements

We thank the NUS Centre for BioImaging Sciences, SingaScope, and the Institute of Molecular and Cell Biology for providing microscope facility support and providing zebrafish care. TW acknowledges funding by the Singapore Ministry of Education (MOE2016-T3-1-005). SV is supported by a NUS Research Scholarship. We thank David Piston for providing the construct of mApple. We thank Tom Carney and Harsha Mahabaleshwar for providing the Tg(*7xTcf*:NLS-mCherry) wnt reporter line.

## Additional information

### Funding

| Funder | Grant reference number | Author |
|---|---|---|
| Ministry of Education - Singapore | MOE2016-T3-1-005 | Thorsten Wohland<br>Paul T Matsudaira |

The funders had no role in study design, data collection and interpretation, or the decision to submit the work for publication.

### Author contributions

Sapthaswaran Veerapathiran, Conceptualization, Resources, Data curation, Formal analysis, Investigation, Methodology, Writing - original draft; Cathleen Teh, Resources, Methodology, Writing - review and editing; Shiwen Zhu, Resources, Software, Writing - review and editing; Indira Kartigayen, Resources, Methodology; Vladimir Korzh, Validation, Writing - review and editing; Paul T Matsudaira, Supervision, Funding acquisition, Writing - review and editing; Thorsten Wohland, Conceptualization, Supervision, Funding acquisition, Validation, Project administration, Writing - review and editing

### Author ORCIDs

Sapthaswaran Veerapathiran (iD) https://orcid.org/0000-0001-5642-1283
Thorsten Wohland (iD) https://orcid.org/0000-0002-0148-4321

### Ethics

Animal experimentation: the study was performed in strict accordance with the Institutional Animal Care and Use Committee (IACUC) protocol of the Biological Resource Center (BRC), A*STAR, Singapore (IACUC #161105) and the National University of Singapore (IACUC# BR18-1023).

### Decision letter and Author response

Decision letter https://doi.org/10.7554/eLife.59489.sa1
Author response https://doi.org/10.7554/eLife.59489.sa2

## Additional files

### Supplementary files

• Transparent reporting form

### Data availability

All data analyzed during this study are included in the manuscript as source files. Source data files have been provided for Figures 4, 5, 6 and Table 1. Videos 1 and 2 represent the raw file used to construct Figures 2-3 and Videos 3-4, respectively.

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
