## [Decision Letter]

**Acceptance summary:**

The movement of morphogen ligands through tissue in vivo is an important and understudied topic. In this study, Veerapathiran et al. examine the sources and distribution mechanism of a morphogen protein, Wnt3, in the developing zebrafish brain. Using a fluorescent Wnt3 fusion protein under the control of Wnt3 regulatory sequences, the authors show that Wnt3 spreads a considerable distance from the cells that produce it, as marked by a membrane-bound fluorescent protein whose production is controlled by the same regulatory elements. The authors also use biophysical methods to characterise the diffusion and mobility of the Wnt fusion protein and to investigate the interaction between Wnt3 and its receptor, Fzd1. This is a high quality study that nicely demonstrates the long range dispersion of Wnt3, and sheds light on the underlying mechanism.

**Decision letter after peer review:**

Thank you for submitting your article "Wnt3 distribution in the zebrafish brain is determined by expression, diffusion and multiple molecular interactions" for consideration by *eLife*. Your article has been reviewed by three peer reviewers, and the evaluation has been overseen by a Reviewing Editor and Marianne Bronner as the Senior Editor. The reviewers have opted to remain anonymous.

The reviewers have discussed the reviews with one another and the Reviewing Editor has drafted this decision to help you prepare a revised submission.

In this study, Veerapthiran et al. examine the sources and spatial distribution of Wnt3 protein in the Zebrafish brain at 24-48hpf using a Wnt-EGFP fusion protein. They also use biophysical methods to characterize the diffusion and mobility of the Wnt fusion and to investigate the interaction between Wnt3 and Frz1. The movement of morphogen ligands through tissue in vivo is an important and understudied topic. This study has the advantage of looking in vivo at proteins expressed at near endogenous levels (the Wnt3 transgene is expressed using the Wnt3 promoter region).

While the reviewers all agreed that the study has potential, they have also identified quite a substantial number of revisions that must be addressed to have confidence in the results and for this study to be of broad interest. Some similar points have come up in all three reviews, which are appended in full below for your information.

Reviewer #1:

1) I am concerned about the comparisons between the source and distribution of Wnt3 using different transgenes with different fluorescent proteins. The data are also presented non-quantitatively so that it is difficult to tell whether, for example, there is mApple fluorescence above background (for source cells) outside of the limited regions described in results. The authors should address: (a) the distribution of Wnt3 mRNA by in situ at 24h (Figure 2—figure supplement 1) appears significantly broader than mApple fluorescence at 24h (b) the source appears to expand considerably from 24 to 48h in mApple fluorescence but there is no corresponding expansion in the Wnt3 mRNA (c) the number of target cells seem to reduce during this time the source is expanding. Data quantifying the distribution of the two fluorescent proteins throughout the region of interest should be shown. It would also be helpful to show in situ data for EGFP so one could compare the source to the target using a single transgene.

2) The authors should better explain the process of model selection. The text simply says that "Bayesian model selection" is applied without explaining further. The authors should explain how their data justify using a 2-component (fast and slow) model, how much is the fit improved by using 2 components compared to 1 component and by including a triplet state? Given the relatively large number of parameters, how do the authors ensure that they avoid overfitting their data? In this regard, it would be helpful for the authors to subject their controls to the analysis with the same model. For example, if secGFP is analyzed with this model, does it yield *F_trip_* ~ 0 and *F_slow_* ~ 0?

3) There isn't sufficient explanation of the biology that the authors believe is behind their measurements. Is the authors' interpretation of the slow fraction that it is bound to the membrane? It is surprising that the slow fraction of Wnt does not differ from that of Lyn which is not secreted? Could this also be Wnt bound to the matrix (e.g. HSPG)? Why does heperanse treatment affect the fast diffusion rate but nothing else? This is supposedly the diffusion of free Wnt. If it is merely clearing more extracellular space, shouldn't the diffusion of secGFP be affected as well? If it is unbinding Wnts from heparin, it would seem that this would be better described by treating bound and unbound Wnts separately.

4) For the measurement of binding affinity, a number is given but the controls are not sufficiently quantitative to give confidence in this number. The best would be to use the same method to calculate a known binding affinity. In the absence of this, is the "binding affinity" computed for the positive control substantially higher than that for Wnt-Frz? In any case, can the authors use this method to spatially map Wnt-Frz interactions in the embryo? This would be more interesting than extracting a binding constant. The authors mention that no binding was detected in the ventral cerebellum or optic tectum but no data is provided.

Reviewer #2:

The authors aim to identify and characterise the expression and mobility of Wnt3 ligands within the developing dorsal midbrain of the zebrafish embryo. To do so, the authors use a previously generated transgenic zebrafish line with fluorescently tagged Wnt3GFP and Fz1mApple under the endogenous Wnt3 promoter Tg(-4.0wnt3:) (The et al., 2015). Using these transgenic lines coupled with other fluorescent probes such as secEGFP, LynEGFP and PMTmApple (membrane-tethered EGFP/mApple) under the same regulatory elements, the authors observed that Wnt3GFP is mobilised from the source regions (roof plate of tectum, midbrain-hindbrain boundary, and dorsal cerebellum) to the distal areas around the tectum and cerebellum. The authors then explored the mode of transport by utilising FCS and FRAP technology. FCS analysis suggested that two components of Wnt3GFP were either fast or slow diffusing. Upon further investigation using the heparin sulphate proteoglycan inhibitor, heparinase, it was observed that the fast component of Wnt3GFP was increased. Still, the slow one was unchanged, suggesting that Wnt3GFP is transported extracellularly with the help of HSPGs.

Furthermore, FRAP measurements confirmed this and proved that the recovery of Wnt3GFP was not as a result of cellular protein renewal. Lastly, the authors quantified the interaction between Wnt3GFP and target cell surface receptor Fzd1mApple. This interaction was consistent with in vitro quantification but at a higher dissociation constant, Kd. Finally, the data suggest an importance of co-receptor Lrp5 and identified that Wnt3GFP-Fz1mApple interaction is strongly dependent on Lrp5 expression.

There are two main points of criticism. Firstly, the authors need to demonstrate the bioactivity of Wnt3GFP in the system. This is very important because it is unclear to which extend Wnt3GFP fulfils the function of Wnt3 during development in comparison to the endogenous Wnt3. Furthermore, the authors need to be more explicit in the text that they describe the characteristics of Wnt3GFP and Fzd1mApple – and not the endogenous protein. This needs to be amended at many places in the text/figure legends etc. if appropriate

Secondly, the studies using FCCS to determine the Kd between Wnt3GFP and Fzd1mApple with/without Lrp5 in vivo is a highlight of the paper. I would suggest interfering with Fzd1, Wnt3 and Lrp5 function to demonstrate similar requirements for these three components in midbrain development. Although some publications are suggesting a role in midbrain development for these components, functional validation of the FCCS experiments would significantly strengthen the second core message of the manuscript.

Conclusion: Overall, this paper adopts a robust approach to investigating the in vivo dynamics of Wnt3GFP in the developing zebrafish brain. This paper delivers novel data with little evident criticism of the experimental approach and conclusions drawn.

Reviewer #3:

This paper focuses on the mechanism whereby Wnt3 is distributed from the producing cells in the developing zebrafish brain. The authors use a Wnt3EFGP fusion protein under the control of Wnt3 regulatory sequences, and show that it spreads considerably away from the producing cells, as marked by a membrane-bound TmApple fluorescent protein whose production is controlled by the same regulatory elements. They use FCS and FRAP to obtain diffusion coefficients, and demonstrate that HSPGs restrain the mobility of Wnt3. They go on to calculate the binding affinity for the frizzled receptor (Fzd1) and show that Wnt3's ability to bind Fzd1 requires Lrp5.

I think that this is a nice study and for the most part the work is well done. I do think though that in order to interpret the data correctly, considerably more controls are needed. There are also some other missing experiments that would strengthen the findings of the study.

1) The first issue concerns the use of the 4 kb upstream region of Wnt3 to control expression of the Wnt3EGFP and the PMTmApple. The authors say that it is a faithful reporter of Wnt3 expression, yet the wholemount in situs shown show a much wider distribution then the PMTmApple would suggest. This made me wonder whether the expression of the Wnt3EGFP and PMTmApple is actually wider than the authors assume, which would confound the results as it could be that the Wnt3EGFP is not as mobile as they think. To control for this, it is important to perform an EGFP in situ to prove that the source of Wnt3EGFP really is localized.

2) The second issue concerns the possibility that free EGFP could be produced by cleavage of the Wnt3EGFP. If this was occurring, then it could contribute to the fast diffusing species measured. The authors must be able to rule this out. A useful way of determining whether this is happening would be a Western blot of Wnt3EGFP-expressing embryos using an antibody against EGFP. This would certainly show whether there was a substantial amount of cleavage, which would be obvious as a band running at the size of EGFP, in addition to a higher band corresponding to the fusion. Moreover, the authors look at the diffusion of injected sec-EGFP. This is an important control, but I think that a better control would be to express the sec-EGFP from the same enhancer that is driving the Wnt3EGFP. It would then be possible to compare exactly the behaviour of the Wnt3 fusion protein with free EGFP.

Another important control would be to show that the Wnt3EGFP distant from the source can induce Wnt signalling as perhaps readout by nuclear β-catenin, or by specific target gene expression.

3) The authors need to provide proof that the heparinase was effective and also that the Lrp5 morpholino was first of all effective, and secondly, specific. They need to show good knockdown of Lrp5, and show that they can rescue the effect with re-expression of Lrp5.

It is also essential to measure the affinity of Lrp5 for Wnt3, as this is crucial to their model.

4) The authors show a big discrepancy in the diffusion coefficients they measure with FCS versus those that they measure with FRAP. This is true of both Wnt3EGFP and also Sec-EGFP, although much more dramatic for Wnt3EGFP. This issue needs resolving with more experiments. For Wnt3EGFP, the authors suggest in the Discussion that this is due to transient binding to receptors and HSPG, but they haven't shown this. This would also affect the FCS measurements, as shown in Table 1 and wouldn't explain the discrepancy for Sec-EGFP. Receptor binding also should influence both the FCS and FRAP measurements. I think that the authors need to explore this in more detail to be certain that the diffusion coefficients that they are measuring with the different approaches are real.

[Editors' note: further revisions were suggested prior to acceptance, as described below.]

Thank you for resubmitting your work entitled "Wnt3 distribution in the zebrafish brain is determined by expression, diffusion and multiple molecular interactions" for further consideration by *eLife*. Your revised article has been evaluated by Marianne Bronner (Senior Editor) and a Reviewing Editor.

The manuscript has been improved but two of the reviewers have highlighted some remaining issues that should be addressed before acceptance, as outlined below. We leave any further experimental work to your discretion, but if further experiments (e.g. fluorescent ISH to determine source cells) are not possible, please revise the text to indicate some uncertainty in your results. The reviewers have also suggested that it would be useful to discuss your results in the light of the recent paper from the JP Vincent lab. Again, this is not an essential revision, but we wanted to give you the opportunity to see these comments and include this discussion if you wish.

Reviewer #1:

The authors have responded satisfactorily to most of my comments and the manuscript is significantly improved. However, I am still not convinced by the authors' response regarding detecting the source and target regions by colocalization analysis. In particular, even in the clearer images provided of the Wnt3-GFP transcription (supplement to Figure 1), the area appears broader than that of the mApple fluorescence. If for whatever reason, the mApple fluorescence was dimmer than the EGFP signal and more difficult to detect, the colocalizaition analysis could miss the regions where mApple is dimmer. The authors indicate that the lower resolution of the in situ hybridizations makes quantification difficult, which is true, however this is not an inherent limitation as it is possible to perform quantitative fluorescent in situ hybridization. This point is central to the manuscript and I think it is important that the authors address it clearly before publication.

Reviewer #2:

I am happy to see that the manuscript has improved and most of my suggestions have been included in the revised manuscript. I have just a few comments:

I can see that the data for the 7xTCF-mCherry reporter, which shows expression overlapping with the Wnt3GFP signal. However, is there a way to demonstrate that Wnt3GFP regulates the activation of the reporter? I am worried that endogenous Wnt3 or another canonical Wnt regulates the expression of the reporter.

I agree, it would be helpful to exclude that there is free GFP in the Wnt3GFP embryos by a Western blot analysis against GFP (as suggested by reviewer 3).

In a recent publication from the JP Vincent lab it has been suggested that Wg binds to Glypicans (McGough et al., 2020). Would this be a possible mechanism to explain distribution of the fast population of Wnt3GFP? After heparinase treatment, a faster recovery rate was observed with an increase of about a factor 2-3 in Wnt3GFP diffusion coefficient. It is unclear to me how a lipid-modified protein – which is solubilized by glypicans – diffuses faster when the HSPGs are reduced. The authors need to discuss their findings in the light of the most recent published data.

The experiments with Surfen is well done and backs up the point made by the heparinase treatment.

Reviewer #3:

This paper focuses on the mechanism whereby Wnt3 is distributed from the producing cells in the developing zebrafish brain. The authors use a Wnt3EFGP fusion protein under the control of Wnt3 regulatory sequences, and show that it spreads considerably away from the producing cells, as marked by a membrane-bound TmApple fluorescent protein whose production is controlled by the same regulatory elements. They use FCS and FRAP to obtain diffusion coefficients, and demonstrate that HSPGs restrain the mobility of Wnt3. They go on to calculate the binding affinity for the frizzled receptor (Fzd1) and show that Wnt3's ability to bind Fzd1 requires Lrp5. This is a high quality study that demonstrates very nicely long range dispersion of Wnt3, and sheds light on the underlying mechanism.

The revised version of the manuscript is substantially improved and the authors have now addressed all my previous concerns and criticisms.

---

## [Author Response]

While the reviewers all agreed that the study has potential, they have also identified quite a substantial number of revisions that must be addressed to have confidence in the results and for this study to be of broad interest. Some similar points have come up in all three reviews, which are appended in full below for your information.Reviewer #1:1) I am concerned about the comparisons between the source and distribution of Wnt3 using different transgenes with different fluorescent proteins. The data are also presented non-quantitatively so that it is difficult to tell whether, for example, there is mApple fluorescence above background (for source cells) outside of the limited regions described in results. The authors should address: (a) the distribution of Wnt3 mRNA by in situ at 24h (Figure 2—figure supplement 1) appears significantly broader than mApple fluorescence at 24h (b) the source appears to expand considerably from 24 to 48h in mApple fluorescence but there is no corresponding expansion in the Wnt3 mRNA (c) the number of target cells seem to reduce during this time the source is expanding. Data quantifying the distribution of the two fluorescent proteins throughout the region of interest should be shown. It would also be helpful to show in situ data for EGFP so one could compare the source to the target using a single transgene.

To avoid discrepancies in identifying the source and target regions of Wnt3, such as masking of fluorescence expression by background, we quantify the overlap and difference in expression of Wnt3EGFP and PMTmApple by colocalization analysis. Through this, we generate a heat map on the extent colocalization between PMT-mApple and Wnt3EGFP expression in different regions at 24 hpf and 48 hpf. This has been shown in Video 1,2 and the method adopted to quantify colocalization has been discussed in Materials and methods. The colocalized pixels and non-colocalized pixels were subsequently reconstructed to identify the source and distal target regions respectively.

a) The background staining and low resolution 2D images of wnt3 transcript expression makes the interpretation of in situ hybridization images to identify Wnt3 source regions challenging. It is for this reason we use quantitative colocalization analysis of Wnt3EGFP and PMTmApple fluorescence signal to identify the source and distal target regions in this article. We have now provided clear in situ images of wnt3 egfp transcripts expression in Figures 2,3 —figure supplement 3, which corresponds to the identified source regions by colocalization analysis. These results are consistent with previously published results (Clements et al., 2009; Duncan et al., 2016; C. Teh et al., 2015). Additionally, we have also included images of the Wnt reporter line 7xTCF-NLS mCherry, which corresponds to the target regions (Figures 2,3—figure supplement 3).

b) and c) – Yes the source regions seems to be expanding with time and the number of distal target regions seem to be reducing. While optic tectum (OT) was a distal target region at 24 hpf, some dorso-lateral regions of OT act as a source at 48 hpf. We have now included this in our Discussion.

2) The authors should better explain the process of model selection. The text simply says that "Bayesian model selection" is applied without explaining further. The authors should explain how their data justify using a 2-component (fast and slow) model, how much is the fit improved by using 2 components compared to 1 component and by including a triplet state? Given the relatively large number of parameters, how do the authors ensure that they avoid overfitting their data? In this regard, it would be helpful for the authors to subject their controls to the analysis with the same model. For example, if secGFP is analyzed with this model, does it yield F_trip_ ~ 0 and F_slow_ ~ 0?

We explain now the model selection with some more detail in the Materials and methods section of the article. We originally only mentioned the Bayes model selection for Wnt3 as we have previously published these results in an article in Analytical Chemistry (Sun et al., 2015). The Bayes model selection provides a means to determine which model is the most likely given the data and its noise. The results showed that the most likely model (> 80%) is a two-component model including a “triplet”. The “triplet” actually represents the photophysics of EGFP that includes potentially more than one component, i.e. triplet state, protonation-deprotonation equilibria, etc. (Widengren et al., 1999). All data in our experiments are fitted with the two-component model including a triplet, except secEGFP. Bayes model selection indicated a one-component model with triplet as the more likely model (> 80%) for secEGFP. For this protein, we therefore recover only a fast diffusive component and a triplet component due to the photophysics of EGFP mentioned above.

To answer the reviewer’s question on using inappropriate models, if we use the 2 component model with one triplet for secEGFP, then the triplet and first component will provide the same values within the margins of error as the one component with triplet does. But in general the second diffusive component will vary widely often providing nonphysical values for the fraction or diffusion coefficient. The problem is that the quality of fit of the data will seem to increase as the data is essentially overfit and too many parameters are used. Thus, one needs to rely on the ad hoc judgement whether the parameters make sense within the framework of the experiments. To avoid this ad hoc judgement, Bayes model selection provides the basis for preferring one model over the others, optimizing quality of fit and avoiding over fitting by using too many parameters.

3) There isn't sufficient explanation of the biology that the authors believe is behind their measurements. Is the authors' interpretation of the slow fraction that it is bound to the membrane? It is surprising that the slow fraction of Wnt does not differ from that of Lyn which is not secreted? Could this also be Wnt bound to the matrix (e.g. HSPG)? Why does heperanse treatment affect the fast diffusion rate but nothing else? This is supposedly the diffusion of free Wnt. If it is merely clearing more extracellular space, shouldn't the diffusion of secGFP be affected as well? If it is unbinding Wnts from heparin, it would seem that this would be better described by treating bound and unbound Wnts separately.

We thank the reviewer for pointing out the omission. We added now more explanations about our interpretation of the diffusion coefficients in the article. The reviewer is correct and the slow diffusion component represents proteins diffusing in the membrane. Membrane diffusion is not strongly dependent on the size of the protein (Singer and Nicolson, 1972; Weiß et al., 2013) and thus proteins even of quite different size exhibit similar diffusion coefficients. In addition, it is more the location and partitioning of membrane proteins within different lipid/protein domains than their size that determines their diffusion. We have shown this also in two earlier articles (Ng et al., 2016; Sezgin et al., 2017). The latter two articles also demonstrate that Wnt3EGFP is influenced by various interventions changing the lipid content of membranes in cells and in vivo. Therefore, the slow component is likely related to the membrane. Wnts are known to travel in the extracellular spaces by constant binding and unbinding to HSPG (Yan and Lin, 2009). Hence, the change in diffusion coefficient upon HSPG cleavage is only seen for the fast component in the extracellular spaces, and not the slow membrane bound component as shown in Table 1 in the manuscript.

If SecEGFP (33 kDa) and Wnt3EGFP (73 kDa) would diffuse freely one would expect a 30% difference in their diffusion coefficient (D is inversely proportional to the cube root of the mass). As Wnt3EGFP interacts with HSPG it is slowed down in its movement. Thus it diffuses a factor ~2 (=57/28) slower than secEGFP. However, after HSPG disruption, secEGFP and Wnt3EGFP differ by a factor of 60/46=1.3 as expected. The increase for diffusion of Wnt3EGFP is thus a result of it not being retarded anymore by fixed HSPG, and not by a reduction of free space.

Finally, free and bound Wnt3EGFP cannot be treated separately. If Wnt3EGFP would be bound for long times, then we would not see bound Wnt3EGFP diffusing at all and would just see photobleaching of the bound species but no diffusion. Any leftover diffusive species would then be diffusing freely. If however, the interactions are transient, then Wnt3EGFP movement would be slowed down due to constantly binding and unbinding to HSPG. This is what we assume to happen here. However, in this case there is no separate treatment as all Wnt3EGFP would oscillate between free diffusion and binding/unbinding. We clarify these points now in the article.

4) For the measurement of binding affinity, a number is given but the controls are not sufficiently quantitative to give confidence in this number. The best would be to use the same method to calculate a known binding affinity. In the absence of this, is the "binding affinity" computed for the positive control substantially higher than that for Wnt-Frz? In any case, can the authors use this method to spatially map Wnt-Frz interactions in the embryo? This would be more interesting than extracting a binding constant. The authors mention that no binding was detected in the ventral cerebellum or optic tectum but no data is provided.

This is an important point. We therefore now mention a number of references in which we have shown that FCCS can quantitatively determine dissociation constants in cells as in vivo, which are consistent with biochemical experiments as far as available (Shi et al., 2009; Sudhaharan et al., 2009; Wang et al., 2016; Yavas et al., 2016). In these articles, we show that we can determine binding affinities ranging from the low nM to the µM range and that these measurements are consistent with measurements in other labs.

For the positive control, we cannot calculate a binding constant as the two fluorophores are permanently linked. The positive control determines the highest possible cross-correlation amplitude that we can obtain and thus is an important control for us whether we see aggregation or bleaching. Aggregation will change the overall fluorescent properties and can lead to cross-correlations much higher than the positive control. Bleaching would lead to cross-correlations whose amplitude would be inconsistent with the amplitudes of the autocorrelation functions. We mention that we did not detect any binding in the ventral cerebellum and optic tectum as we did not detect any cross-correlations between Wnt3EGFP and Fzd1mApple in these regions. We have now included this data in Figure 6—figure supplement 2 A.

Reviewer #2:The authors aim to identify and characterise the expression and mobility of Wnt3 ligands within the developing dorsal midbrain of the zebrafish embryo. To do so, the authors use a previously generated transgenic zebrafish line with fluorescently tagged Wnt3GFP and Fz1mApple under the endogenous Wnt3 promoter Tg(-4.0wnt3:) (The et al., 2015). Using these transgenic lines coupled with other fluorescent probes such as secEGFP, LynEGFP and PMTmApple (membrane-tethered EGFP/mApple) under the same regulatory elements, the authors observed that Wnt3GFP is mobilised from the source regions (roof plate of tectum, midbrain-hindbrain boundary, and dorsal cerebellum) to the distal areas around the tectum and cerebellum. The authors then explored the mode of transport by utilising FCS and FRAP technology. FCS analysis suggested that two components of Wnt3GFP were either fast or slow diffusing. Upon further investigation using the heparin sulphate proteoglycan inhibitor, heparinase, it was observed that the fast component of Wnt3GFP was increased. Still, the slow one was unchanged, suggesting that Wnt3GFP is transported extracellularly with the help of HSPGs.Furthermore, FRAP measurements confirmed this and proved that the recovery of Wnt3GFP was not as a result of cellular protein renewal. Lastly, the authors quantified the interaction between Wnt3GFP and target cell surface receptor Fzd1mApple. This interaction was consistent with in vitro quantification but at a higher dissociation constant, Kd. Finally, the data suggest an importance of co-receptor Lrp5 and identified that Wnt3GFP-Fz1mApple interaction is strongly dependent on Lrp5 expression.There are two main points of criticism. Firstly, the authors need to demonstrate the bioactivity of Wnt3GFP in the system. This is very important because it is unclear to which extend Wnt3GFP fulfils the function of Wnt3 during development in comparison to the endogenous Wnt3. Furthermore, the authors need to be more explicit in the text that they describe the characteristics of Wnt3GFP and Fzd1mApple – and not the endogenous protein. This needs to be amended at many places in the text/figure legends etc. if appropriate

Thank you for pointing this out. We have shown Wnt3EGFP activity in our earlier publication (Cathleen Teh et al., 2015) and also mention this in our manuscript. We also amended now the article to make clear we describe the FP labelled proteins not endogenous ones.

Secondly, the studies using FCCS to determine the Kd between Wnt3GFP and Fzd1mApple with/without Lrp5 in vivo is a highlight of the paper. I would suggest interfering with Fzd1, Wnt3 and Lrp5 function to demonstrate similar requirements for these three components in midbrain development. Although some publications are suggesting a role in midbrain development for these components, functional validation of the FCCS experiments would significantly strengthen the second core message of the manuscript.

In this study, when the interaction of Wnt3 and Fzd1 was perturbed by knocking-down the expression of co-receptor, we observed defective midbrain and hindbrain development (Figure 6—figure supplement 2). Additionally, we also show that the Wnt3-Fzd1-Lrp5 interaction is also important to modulate the distribution of Wnt3EGFP (Figure 6—figure supplement 3). These results show that interaction of Wnt3-Fzd1-Lrp5 is crucial for zebrafish brain development.

Conclusion: Overall, this paper adopts a robust approach to investigating the in vivo dynamics of Wnt3GFP in the developing zebrafish brain. This paper delivers novel data with little evident criticism of the experimental approach and conclusions drawn.

We thank the reviewer for the comments. We have now included additional discussions on the techniques used and the interpretation of the results. We have also included additional control experiments in our revised manuscript to support our conclusions.

Reviewer #3:This paper focuses on the mechanism whereby Wnt3 is distributed from the producing cells in the developing zebrafish brain. The authors use a Wnt3EFGP fusion protein under the control of Wnt3 regulatory sequences, and show that it spreads considerably away from the producing cells, as marked by a membrane-bound TmApple fluorescent protein whose production is controlled by the same regulatory elements. They use FCS and FRAP to obtain diffusion coefficients, and demonstrate that HSPGs restrain the mobility of Wnt3. They go on to calculate the binding affinity for the frizzled receptor (Fzd1) and show that Wnt3's ability to bind Fzd1 requires Lrp5.I think that this is a nice study and for the most part the work is well done. I do think though that in order to interpret the data correctly, considerably more controls are needed. There are also some other missing experiments that would strengthen the findings of the study.1) The first issue concerns the use of the 4 kb upstream region of Wnt3 to control expression of the Wnt3EGFP and the PMTmApple. The authors say that it is a faithful reporter of Wnt3 expression, yet the wholemount in situs shown show a much wider distribution then the PMTmApple would suggest. This made me wonder whether the expression of the Wnt3EGFP and PMTmApple is actually wider than the authors assume, which would confound the results as it could be that the Wnt3EGFP is not as mobile as they think. To control for this, it is important to perform an EGFP in situ to prove that the source of Wnt3EGFP really is localized.

We have previously shown that in situ expression of egfp transcripts under 4kb wnt3 promoter faithfully reports the wnt3 transcripts expression by imaging cross-sections of different regions of zebrafish brain (C. Teh et al., 2015). This is consistent with the expression of wnt3 transcripts published by other groups (Clements et al., 2009; Duncan et al., 2016). We have also now included the expression of egfp transcripts in Figure 1—figure supplement 1 B,D. Additionally, to validate our distal target regions, we image now a Wnt reporter line, 7xTCF-NLS mCherry, which shows nuclear localized mCherry expression in the target regions we identified. We now include these images in Figure 1—figure supplement 1E,F. However, it is important to note that background signal, overstaining and specificity of riboprobes makes identification of the precise Wnt3 source regions challenging. Moreover, it is difficult to distinguish between the autocrine and paracrine target regions from the Wnt reporter line. Therefore, in this article, we use quantitative co-localization analysis of fluorescence signal from Wnt3EGFP and PMTmApple to identify the source and distal target regions. The sensitivity of fluorescence is also quite high and we therefore think that the quantitative colocalization of fluorescent expressing reporters provide an accurate localization.

Regarding the mobility of Wnt3EGFP, this is exclusively determined by FCS and FRAP. Both methods rely on temporal changes of the fluorescence signal and not on pure spatial distributions. Thus, even if the distribution would suffer from imprecision, the temporal changes would not be influenced by that except of a changing background.

2) The second issue concerns the possibility that free EGFP could be produced by cleavage of the Wnt3EGFP. If this was occurring, then it could contribute to the fast diffusing species measured. The authors must be able to rule this out. A useful way of determining whether this is happening would be a Western blot of Wnt3EGFP-expressing embryos using an antibody against EGFP. This would certainly show whether there was a substantial amount of cleavage, which would be obvious as a band running at the size of EGFP, in addition to a higher band corresponding to the fusion. Moreover, the authors look at the diffusion of injected sec-EGFP. This is an important control, but I think that a better control would be to express the sec-EGFP from the same enhancer that is driving the Wnt3EGFP. It would then be possible to compare exactly the behaviour of the Wnt3 fusion protein with free EGFP.

We clearly see that the diffusion coefficient of Wnt3EGFP, obtained by FCS and FRAP, is much lower than secEGFP in the extracellular spaces as well as in the brain ventricle (Figure 4E, Figure 5 and Figure 5—figure supplement 2). Additionally, we see that Wnt3EGFP interacts with HSPG and with Fzd1mApple while secEGFP does not. These results indicate that we are measuring Wnt3EGFP and not free EGFP cleaved from Wnt3EGFP.

We have now performed FCS and FRAP measurements for secEGFP expressed under the 4 kb Wnt3 promoter and shown that we do not see any differences in the dynamics of secEGFP (Figure 4—figure supplement 1).

Another important control would be to show that the Wnt3EGFP distant from the source can induce Wnt signalling as perhaps readout by nuclear β-catenin, or by specific target gene expression.

This is an important point and we’d like to thank the reviewer for the suggestion. We have now imaged the Wnt reporter line 7xTCF-NLS mCherry and show that the nuclear localized mCherry expression corresponds to the regions we identify as the Wnt3 target regions (Figure 1—figure supplement 1 E,F).

3) The authors need to provide proof that the heparinase was effective and also that the Lrp5 morpholino was first of all effective, and secondly, specific. They need to show good knockdown of Lrp5, and show that they can rescue the effect with re-expression of Lrp5.

To validate our heparinase measurements, we repeated our experiments with surfen, a quinolone based derivative, known to exhibit heparin-neutralizing activity and antagonize heparan sulfate – protein interactions (Naini et al., 2018). We show that both heparinase and surfen treatments provide similar results – increase the D_fast_ for Wnt3EGFP by a factor ~ 2 (Table 1). Additionally we provide images of Heparinase and surfen treated Wnt3EGFP expressing embryos showing clusters of Wnt3EGFP due to disruption of HSPG (Table 1—source data 1).

Lrp5: The efficacy of the morpholinos used in this study and the phenotypes associated with it have been characterized in (Willems et al., 2015). Upon treatment with lrp5 MO, we show that it results in deformed midbrain and hindbrain and that FCCS is abolished between Wnt3EGFP and Fzd1mApple. As control, we used a lrp5 mismatch morpholino, which did not abolish the Wnt3-Fzd1 interactions and lead to a deformed brain. We include this in Figure 6—figure supplement 2.

It is also essential to measure the affinity of Lrp5 for Wnt3, as this is crucial to their model.

This is a very interesting point and it will be important to measure Lrp5/Wnt3 interactions. We are working on this but this is beyond the scope of this article as this will also require measurements of interactions of Lrp5 with Fzd1 and the determination of the binding sequence.

4) The authors show a big discrepancy in the diffusion coefficients they measure with FCS versus those that they measure with FRAP. This is true of both Wnt3EGFP and also Sec-EGFP, although much more dramatic for Wnt3EGFP. This issue needs resolving with more experiments. For Wnt3EGFP, the authors suggest in the Discussion that this is due to transient binding to receptors and HSPG, but they haven't shown this. This would also affect the FCS measurements, as shown in Table 1 and wouldn't explain the discrepancy for Sec-EGFP. Receptor binding also should influence both the FCS and FRAP measurements. I think that the authors need to explore this in more detail to be certain that the diffusion coefficients that they are measuring with the different approaches are real.

We are sorry that the manuscript has been unclear about this point. FCS measures the local diffusion coefficient (D_local_) of a molecule in a small observation volume (~ femtoliter range) whereas FRAP measures the global diffusion coefficient (D_global_) of a molecule across large area (~ tens of micrometres). For Wnt3EGFP we find that D_local_ measured from FCS is 27.6 ± 3.9 µm^2^/s whereas D_global_ measured from FRAP is 0.5-1 µm^2^/s. In this article, we investigate the factors that reduces D_global_ 40-100 times when compared with D_local_. In Table 1 and Figure 5—figure supplement 2, we show that binding of Wnt3EGFP to HSPG slows its diffusion by a factor ~2. In Figure 6—figure supplement 3, we show that interaction of Wnt3 with its receptor and co-receptor slows its diffusion by a factor ~5-6. For secEGFP, we observe that D_global_ (12-15 µm^2^/s) is 3-5 times slower when compared to its D_local_ (55 – 60 µm^2^/s) (Figure 4 and Figure 5—figure supplement 3). As secEGFP is only a secreted control that diffuses in the same tissue environment as Wnt3 but does not interact with HSPG or receptors, tortuosity, i.e. the reduction in diffusion due to the presence of cells as obstacles, reduces diffusion of molecules in the interstitial spaces by a factor ~ 3-5. Collectively our results show that long range diffusion of Wnt3EGFP is retarded by a factor 3-5 due to tortuosity, a factor 5-6 due to receptor binding and a factor ∼2 due to HSPG, thus leading to a total reduction of a factor 30-60. These factors alone would be sufficient to explain the discrepancy between FCS and FRAP. We have now explained this more clearly in our article.

[Editors' note: further revisions were suggested prior to acceptance, as described below.]

Reviewer #1:The authors have responded satisfactorily to most of my comments and the manuscript is significantly improved. However, I am still not convinced by the authors' response regarding detecting the source and target regions by colocalization analysis. In particular, even in the clearer images provided of the Wnt3-GFP transcription (supplement to Figure 1), the area appears broader than that of the mApple fluorescence. If for whatever reason, the mApple fluorescence was dimmer than the EGFP signal and more difficult to detect, the colocalizaition analysis could miss the regions where mApple is dimmer. The authors indicate that the lower resolution of the in situ hybridizations makes quantification difficult, which is true, however this is not an inherent limitation as it is possible to perform quantitative fluorescent in situ hybridization. This point is central to the manuscript and I think it is important that the authors address it clearly before publication.

We are pleased that the reviewer found our revised manuscript improved. The reviewer is correct that if for some reason mApple fluorescence intensity were weak and not detected by confocal imaging, this would underestimate the extent of the Wnt3 source regions. Therefore, in the double transgenic line expressing Wnt3EGFP and PMTmApple, we performed FCS measurements in the regions where Wnt3EGFP but not PMTmApple was detected by confocal imaging (optic tectum and ventral regions). The expression of Wnt3EGFP was used as a guide to ensure FCS measurements were performed on the cell membrane. As FCS is a single molecule sensitive technique, it should detect any low intensity fluctuations present in these regions not captured by confocal imaging. As shown in Figure 2—figure supplement 1 and Figure 3—figure supplement 1, we obtained autocorrelation functions for Wnt3EGFP in these regions but not for PMTmApple, indicating no weak mApple signal were present in these regions. Additionally, we also imaged another reporter line driven by the same 4 kb wnt3 promoter, Tg(4.0wnt3:EGFP), expressing cytosolic EGFP in the domains of endogenous wnt3 (Figure 2—figure supplement 2 and Figure 3—figure supplement 3). The expression pattern of EGFP in Tg(-4.0wnt3:EGFP) closely resembles the expression of PMTmApple in Tg(-4.0wnt3:PMTmApple), further suggesting that mApple intensity was not dimmer than EGFP signal in those regions. However, as the reviewer suggested, performing quantitative fluorescence in situ hybridization will accurately recognize the full range and precise boundaries of Wnt3 source and target regions, but this is beyond the scope of this article. Hence, the full range of source and target regions might be longer than what we had identified. Nevertheless, our results indicates the presence of discrete Wnt3 producing- and receiving-regions in the developing brain of zebrafish embryos. We have now included this in our Discussion section (see paragraph three in Discussion section).

Reviewer #2:I am happy to see that the manuscript has improved and most of my suggestions have been included in the revised manuscript. I have just a few comments:

We are pleased that the reviewer found our revised manuscript improved.

I can see that the data for the 7xTCF-mCherry reporter, which shows expression overlapping with the Wnt3GFP signal. However, is there a way to demonstrate that Wnt3GFP regulates the activation of the reporter? I am worried that endogenous Wnt3 or another canonical Wnt regulates the expression of the reporter.

The expression of nuclear localized mCherry in 7xTCF-mCherry reporter line is not regulated by Wnt3EGFP but by endogenous Wnt ligands. This reporter line was only used to show that downstream Wnt signaling is activated in the regions distant from the source that we classified as target regions in Figures 2 and 3. We now clarify this in our manuscript (Discussion section, fourth paragraph).

I agree, it would be helpful to exclude that there is free GFP in the Wnt3GFP embryos by a Western blot analysis against GFP (as suggested by reviewer3).

As clarified in our answer to point 2 by reviewer 3 earlier, we see that the diffusion coefficient of Wnt3EGFP, obtained by FCS and FRAP, is much lower than that of secEGFP in the extracellular spaces as well as in the brain ventricle (Figure 4E, Figure 5 and Figure 5—figure supplement 2). Additionally Wnt3EGFP binds to HSPG and interacts with Fzd1mApple while secEGFP does not. These results indicate that we are measuring Wnt3EGFP and not free EGFP cleaved from Wnt3EGFP.

In a recent publication from the JP Vincent lab it has been suggested that Wg binds to Glypicans (McGough et al., 2020). Would this be a possible mechanism to explain distribution of the fast population of Wnt3GFP? After heparinase treatment, a faster recovery rate was observed with an increase of about a factor 2-3 in Wnt3GFP diffusion coefficient. It is unclear to me how a lipid-modified protein – which is solubilized by glypicans – diffuses faster when the HSPGs are reduced. The authors need to discuss their findings in the light of the most recent published data.

The increase in the diffusion coefficient of the fast component by a factor ~ 2 after disrupting HSPG suggests that Wnt3 spreads by extracellular diffusion and that binding of Wnt3 to HSPG regulates its diffusion. As Wnts are known to travel in the extracellular spaces by transient binding and unbinding to HSPG (Yan and Lin, 2009), the increase in diffusion coefficient upon HSPG cleavage is only seen for the fast component which is diffusing in the extracellular spaces. Our results are in line with the recent findings of JP Vincent lab (McGough et al., 2020) who also reported that Wnt ligands travel long distance by interacting with glypicans, a major family of HSPG, through its lipid moiety. This is indeed a possible mechanism that explains the spreading of the fast component of Wnt3EGFP. Collectively, these results imply that the extracellular diffusion of Wnt ligands are regulated by HSPG. We have now included this in the Discussion section.

**References**

Singer, S. J., & Nicolson, G. L. (1972). The fluid mosaic model of the structure of cell membranes. *Science*, *175*(4023), 720–731.Weiß, K., Neef, A., Van, Q., Kramer, S., Gregor, I., & Enderlein, J. (2013). Quantifying the diffusion of membrane proteins and peptides in black lipid membranes with 2-focus fluorescence correlation spectroscopy. *Biophysical Journal*, *105*(2), 455–462.Widengren, J., Mets, Ü., & Rigler, R. (1999). Photodynamic properties of green fluorescent proteins investigated by fluorescence correlation spectroscopy. *Chemical Physics*, *250*(2), 171–186.